# CaLMFlow: Volterra Flow Matching using Causal Language Models

## Abstract

We introduce CaLMFlow (Causal Language Models for Flow Matching), a novel framework that casts flow matching as a Volterra integral equation (VIE), leveraging the power of large language models (LLMs) for continuous data generation. CaLMFlow enables the direct application of LLMs to learn complex flows by formulating flow matching as a sequence modeling task, bridging discrete language modeling and continuous generative modeling. Our method implements tokenization across space and time, thereby solving a VIE over these domains. This approach enables efficient handling of high-dimensional data and outperforms ODE solver-dependent methods like conditional flow matching (CFM). We demonstrate CaLMFlow's effectiveness on synthetic and real-world data, including single-cell perturbation response prediction, showcasing its ability to incorporate textual context and generalize to unseen conditions. Our results highlight LLM-driven flow matching as a promising paradigm in generative modeling, offering improved scalability, flexibility, and context-awareness.

## 1 Introduction

Recent advances in deep learning have revolutionized generative modeling for complex, high-dimensional data. In particular, methods based on ordinary differential equations (ODEs), such as continuous normalizing flows (CNFs) (Chen et al., 2018) and flow matching (Lipman et al., 2022), have emerged as efficient tools designed for modeling continuous data distributions. However, many ODE systems suffer from stiffness making them numerically unstable and computationally expensive to solve accurately (Kushnir & Rokhlin, 2012; Zappala et al., 2024). In contrast, integral equations (IEs) offer a more generalized framework for capturing dynamics, with IE solvers demonstrating greater numerical stability than their ODE counterparts (Kushnir & Rokhlin, 2012; Zappala et al., 2024). Recent work in operator learning (Xiong et al., 2021; Cao, 2021; Zappala et al., 2024) has also connected solving integral equations with transformers, the foundational architecture of large language models (LLMs), inspiring the use of LLMs to model dynamical systems through the lens of IEs.

Building on these insights, our work introduces **Causal Language Models for Flow Matching (CaLMFlow)**, a novel approach that models flow matching using Volterra integral equations (Zappala et al., 2023), enabling learning flows in a more robust manner. By leveraging causal language models (CLMs) to solve Volterra integral equations, our method capitalizes on the ability of CLMs to comprehend natural language, allowing for the modeling of complex data distributions conditioned on natural language prompts. Our approach provides a robust framework for applications ranging from synthetic data generation to complex system modeling in the biological sciences.

Our key contributions are:

- **Flow matching using causal language models:** We introduce a novel framework formulating flow matching as Volterra integral equations and leveraging causal language models (CLMs) to approximate the solutions, enhancing stability and performance of flow matching.

- **Controllable generation of flows using natural language:** We present a flexible and effective approach to controllable generation conditioned on textual prompts by leveraging the causal language model's natural language understanding. In Section 5.2, we demon-

Figure 1: Overview of the CaLMFlow framework. CaLMFlow takes as input textual conditions and flows and generates the next time point for the flows. The textual condition is tokenized and embedded using the LLM tokenizer and embedding layer while the conditional flows are transformed into spatial-temporal tokens using a learned projection. If multiple conditional flows are input simultaneously, the tokens are ordered by flow, space, and then time. The LLM applies causal language modeling and generates the next time point for each flow.

strate its superiority by applying it to perturbation response prediction in single-cell data, outperforming traditional flow matching and popular single-cell methods.

- **Continuous space tokens via variational decoding:** We introduce variational decoding to sample and generate continuous data. This approach models a continuous conditional distribution for next-token sampling, extending language modeling techniques, which are designed to model discrete data such as texts, to continuous domains. Our ablation study in Section 6.1 demonstrate that variational decoding is crucial for accurately modeling continuous data within our framework.

- **Spatiotemporal and trajectory tokenization:** We present a spatiotemporal tokenization scheme that enables CaLMFlow to model VIEs across both spatial and temporal domains. Additionally, by modeling multiple flows concurrently, CaLMFlow captures correlations between data samples. We demonstrate in Sections 5.1 and 6.3 this approach significantly improves performance.

## 2 RELATED WORK

**Flow Matching and Continuous Normalizing Flows:** Flow matching has significantly enhanced the efficiency and scope of continuous normalizing flows (CNFs) (Chen et al., 2018; Papamakarios et al., 2021) in modeling continuous data distributions. Conditional Flow Matching (CFM) (Lipman et al., 2022) allows for precise control over the generative process by optimizing conditional vector fields tailored for specific distribution paths, including those based on Gaussian distributions. Tong et al. (2024) generalized the conditional paths and introduced mini-batch optimal transport and Schrödinger bridge CFM, improving the efficiency and performance of CFM models. In Hu et al. (2024), flow matching is applied to text generation in a non-autoregressive manner, showing improvements compared to other diffusion-based text generation models such as DiffSeq (Gong et al., 2023). Our work, however, is primarily concerned with adapting LLMs to generate continuous data conditioned on text.

**Text-conditional Generation:** Text-conditional image generation has made significant strides through the integration of diffusion models (Sohl-Dickstein et al., 2015; Ho et al., 2020; Song et al., 2021) and large language models (LLMs). State-of-the-art systems like Stable Diffusion (Rombach et al., 2022), DALLE-2 (Ramesh et al., 2022), and DINOv2 (Oquab et al., 2024) leverage LLM embeddings to generate high-quality images from textual descriptions. Recent research (Ding et al., 2021; Yu et al., 2022; Ge et al., 2024; Zhan et al., 2024) has focused on adapting LLMs for multimodal generation, often employing vector quantization (van den Oord et al., 2017; Razavi et al., 2019; Ge et al., 2023) to extend LLM vocabularies with latent tokens representing non-textual data. Our CaLMFlow method introduces a novel approach as the first flow matching-based text-

conditional generative model that produces continuous tokens, potentially offering greater flexibility and expressiveness compared to discrete token-based methods.

**Integral Equations in Neural Network Frameworks:** The fusion of neural networks and differential equations was introduced by Chen et al. (2018) with neural ordinary differential equations. This concept has since been extended to integral equations, particularly Volterra equations, in several studies (Fu & Hirsa, 2022; Zappala et al., 2023; 2024). Notably, Zappala et al. (2024) exploited the relationship between attention mechanisms and integral kernels (Xiong et al., 2021; Cao, 2021) to model integral equations using transformers. Concurrently, approaches like physics-informed neural networks (PINNs) (Raissi et al., 2019; Lu et al., 2021; Goswami et al., 2022) have emerged, incorporating physical laws as prior knowledge to enhance model accuracy and generalization in operator learning tasks. Our work builds upon these advancements, extending the application of Volterra integral equations to the flow matching framework.

## 3 VOLTERRA FLOW MATCHING

### 3.1 FLOW MATCHING AS VOLTERRA INTEGRAL EQUATIONS

Flow matching (Lipman et al., 2022) is typically formulated as learning the time-dependent vector field $v(x, t)$ generating the *flow* $\phi(x, t)$ related by the ordinary differential equation (ODE):

$$\frac{d\phi}{dt} = v(\phi, t), \qquad \phi(x, 0) = x, \tag{1}$$

transforming an initial distribution $p_0$, usually Gaussian noise, at time $t = 0$ into any target distribution $p_1$ at time $t = 1$ through the application of a numerical ODE solver applied to the learned vector field, resulting in a gnerative model of distribution $p_1$ However, ODEs can suffer from stiffness, especially when modeling systems with rapid changes, long-range dependencies or in high dimensions (Rokhlin, 1985; 1990; Kushnir & Rokhlin, 2012; Zappala et al., 2024) and, as a result, solving such systems are highly numerically unstable and computationally expensive. To address these challenges fundamental to ODE systems, we transform Equation 1 into its equivalent integral form by integrating over time:

$$\phi(t) = \phi_0 + \int_0^t v(\phi(s), s)ds. \tag{2}$$

This is a Volterra integral equation (VIE) of the second kind for the unknown function $\phi$ and $v$ is a kernel function. The VIE formulation generalizes the ODE approach and offers several advantages: it inherently accounts for nonlocal components in the dynamics, is more flexible and robust for modeling complex systems with memory effects, and avoids issues like stiffness and underflowing that ODE solvers encounter (Kushnir & Rokhlin, 2012; Zappala et al., 2024). Consequently, Volterra flow matching provides a more general and stable approach to modeling continuous flows between distributions.

### 3.2 SOLVING VOLTERRA INTEGRAL EQUATIONS WITH CAUSAL LANGUAGE MODELS

In CaLMFlow, we define the flow using a more general form of Equation 2:

$$z_t = f(z_t, t) + \int_0^t G(z_s, t, s)ds, \tag{3}$$

where $z(t)$ is the underlying flow and we use shorthand notation $z_t$ for simplicity. The term $f(z_t, t)$ serves as an inhomogeneous component in a general VIE that encodes local changes in the system and the integral $\int_0^t G(z_s, t, s)ds$ captures the accumulated influence of the dynamics over the history.

Following prior works Xiong et al. (2021); Cao (2021); Zappala et al. (2024), we utilize a causal language model (CLM) to model the underlying dynamics. We discretize the time domain $[0, 1]$ into $N$ time steps, denoted as $(t_0, t_1, \ldots, t_N)$, where $t_0 = 0$ and $t_N = 1$. Discretized conditional flow trajectories are then sampled at these time points from a given ground truth continuous flow trajectory $z(t)$ (the choice of $z(t)$ will be discussed later). This process yields the sequence $(z_{t_0}, z_{t_1}, \ldots, z_{t_N})$, which serves as input to the CLM. Intuitively, the CLM employs its attention mechanism to encode information from $z_0$ through $z_t$ and predict $z_{t+1}$.

This process can be formalized as a sequence of functions $\left(z^0(t), z^1(t), \ldots, z^N(t)\right)$, where each $z^i(t)$ is defined on the domain $\{t_0, t_1, \ldots, t_i\}$ and satisfies $z^i(t_j) = z_{t_j}$. Superscripts indicate functions of time $t$ (representing discretized flow trajectories), while subscripts denote the values of these functions at specific time points. For simplicity and alignment with sequence-based CLM inputs, we represent $z^i(t)$ as the sequence of its values $(z_0, \ldots, z_i)$ over its domain. The CLM then acts as an iterative solver for the discretized Volterra integral equation (VIE). It predicts $\hat{z}^{i+1}$ from $z^i$, extending the domain to the next time step:

$$\hat{z}^{i+1} = f(z^i, t_{i+1}) + \sum_{j=0}^{i} \Delta t_{i+1} G(z_j, t_{i+1}, t_j), \tag{4}$$

where $\Delta t_k = t_k - t_{k-1}$.

The iterative solving process is performed in parallel using teacher forcing, where the CLM is trained on ground-truth discretized trajectories. The model's final output $z^N$ is the "solved" trajectory.

Further theoretical discussion, framed in the context of Banach spaces, is provided in Appendix D. Generalizations to higher-dimensional integrals, where spatial dependencies are also considered, are straightforward but omitted for clarity.

Similar to the discussion in Lipman et al. (2022), the naïve Volterra flow matching objective

$$\mathcal{L}_{\text{VFM}} = \mathbb{E}_{p(z^N)} \left\| z^N - \hat{z}^N \right\|^2 = \mathbb{E}_{p(z^N)} \left[ \sum_{i=0}^{N} \left\| z_{t_i} - \hat{z_{t_i}} \right\|^2 \right], \tag{5}$$

where $z^N$ is the ground truth flow trajectory (a function defined on the discretized time grid $\{t_0, \ldots, t_N\}$), $\hat{z}^N$ is the model predicted trajectory and $p(z^N)$ is the distribution of trajectories over the flow. However, $p(z^N)$ is intractable due to the inaccessibility of ground truth marginal trajectories $z^N$. Instead, we use the optimal transport conditional probability paths (Lipman et al., 2022; Tong et al., 2024), where we sample each $z_{t_j}$ of the conditional flow trajectory $z^N$ from $p_{t_j}(z|z_0, z_1) = \mathcal{N}(z|(1-t_j)z_0 + t_j z_1, \sigma_{t_j})$, where $z_0$ and $z_1$ are sampled from the source and target distributions as conditions.

We optimize the conditional Volterra flow matching objective

$$\mathcal{L}_{\text{CVFM}} = \mathbb{E}_{p_0(z_0), q(z_N)} \left\| z_{z_0, z_N}^N - \hat{z}^N \right\|^2, \tag{6}$$

where $p_0$ is the initial source distribution (e.g., a Gaussian) and $q$ is the target data distribution. A theoretical discussion of the different objective functions can be found in Appendix E. We note that next-token prediction in CLMs is simulation-free during training, as the next token is predicted based on the ground truth history. Full trajectory simulation occurs only during inference. As such, CaLMFlow, like CFM, operates as a simulation-free approach during training.

### 3.3 Continuous Space Tokens via Variational Decoding

We introduce variational decoding with a Kullback-Leibler divergence regularizer in our experiments. Unlike standard CLMs, which rely on a fixed vocabulary and model the next-token distribution through softmax probabilities, our approach enables the CLM to model a continuous distribution of next tokens. Specifically, we use a probabilistic encoder $q_\phi(\mathbf{z}|\mathbf{x})$ to map the CLM output tokens $\mathbf{x}$ to a posterior latent distribution $\mathcal{N}(\mathbf{z}; \boldsymbol{\mu}, \boldsymbol{\sigma}^2\mathbf{I})$, and a probabilistic decoder $p_\psi(\mathbf{x}|\mathbf{z})$ to reconstruct the tokens $\mathbf{x}$, where latent variable $\mathbf{z}$ acts as a continuous representation in a latent space.

Both $q_\phi$ and $p_\psi$ are optimized by maximizing the evidence lower bound (ELBO) (Kingma & Welling, 2022):

$$-\mathcal{L}_{\text{VAE}} = \mathbb{E}_{q_\phi(\mathbf{z}|\mathbf{x})}[\log p_\psi(\mathbf{x}|\mathbf{z})] - \beta \, \text{KL}(q_\phi(\mathbf{z}|\mathbf{x})\|p(\mathbf{z})), \tag{7}$$

where $p(\mathbf{z}) := \mathcal{N}(\mathbf{z}; \mathbf{0}, \mathbf{I})$ is a prior over the latent variable $\mathbf{z}$, $\beta$ is a scaling hyperparameter as introduced in (Higgins et al., 2017), and $\text{KL}(q\|p)$ is the Kullback-Leibler divergence. Using the ELBO, our Volterra flow matching objective becomes

$$\mathcal{L}_{\text{VCVFM}} = \mathcal{L}_{\text{CVFM}} + \beta \, \text{KL}(q_\phi(\mathbf{z}|\mathbf{x})\|p(\mathbf{z})) \tag{8}$$

To control the generation of continuous tokens at inference, we use a temperature parameter $\tau$ that scales the variance of the encoded posterior, analogous to the use of temperature in LLMs (Renze & Guven, 2024; Peeperkorn et al., 2024). Specifically, the posterior latent distribution is modified as $q_\phi(\mathbf{z}|\mathbf{x}) \sim \mathcal{N}(\mathbf{z}; \boldsymbol{\mu}, \tau\boldsymbol{\sigma}^2\mathbf{I})$, where $\tau \geq 0$ is the temperature parameter. An ablation experiment for the temperature parameter is in Section 6.1.

# 4 SPATIOTEMPORAL AND MULTI-TRAJECTORY TOKENIZATION

In Section 3.2, we discussed how causal language models (CLMs) are utilized to solve integral equations by discretizing the time domain. In this section, we first extend the integration to include spatial domains, enabling the CLM to simultaneously model dynamics over both space and time. Subsequently, we introduce a heuristic method that encodes multiple flow trajectories within the same input sequence, which is empirically proven to improve model performance. For precise implementation details during training and inference, see Algorithms 1 and 2 in Appendix B.

## 4.1 SPATIOTEMPORAL TOKENIZATION

As discussed in Section 3.2, we first discretize the temporal domain $[0, 1]$ using a fixed grid $\{t_i\}_{i=1}^N$ of length $N$. The discretized temporal dynamics are encoded as input matrix $\mathbf{Z} \in \mathbb{R}^{T \times D_{in}}$ comprising T tokens, where each token $\mathbf{z}_i$ is of $D_{in}$ dimensions. To introduce spatial structure into the representation, we further subdivide each temporal embedding vector $\mathbf{z}_i \in \mathbb{R}^{D_{in}}$ into $K$ vectors $\{\mathbf{z}_{ij}\}_{j=1}^K$. The splitting can be either learned or predefined, depending on the data. For instance, for single-cell data, we map the encoded gene expression features with a learned network $S_\theta : \mathbb{R}^{D_{in}} \to \mathbb{R}^{K \cdot D_{in}}$ and split the token to $K$ spatial tokens. In the case of images, we use an encoder to extract low-level features. By ordering the embedding vectors according first to the time step and then the spatial order, the final spatiotemporal token sequence $\mathbf{Z}_{st} \in \mathbb{R}^{T \cdot K \times D_{in}}$ is represented as $\mathbf{Z}_{st} := [\mathbf{z}_{11}, \ldots, \mathbf{z}_{1K}, \ldots, \mathbf{z}_{N1}, \ldots, \mathbf{z}_{NK}]^\top$. This spatiotemporal sequence of tokens is processed by the CLM, which approximates a Volterra integral equation over both space and time. The discussion in Section 3.2 extends to a spatiotemporal VIE.

## 4.2 MULTI-TRAJECTORY TOKENIZATION

We embed more than one conditional trajectory as input to the CLM by sampling $M$ spatiotemporal sequences as above and concatenate them along the sequence dimension. The overall sequence $\mathbf{Z}_{sdt} \in \mathbb{R}^{M \cdot K \cdot D_{in}}$ becomes $\mathbf{Z}_{sdt} := [\mathbf{z}_{111}, \ldots, \mathbf{z}_{1K1}, \ldots, \mathbf{z}_{1KM}, \ldots, \mathbf{z}_{N11}, \ldots, \mathbf{z}_{NKM}]^\top$.

We empirically find that providing information across a batch of trajectories as context benefits model performance. We observe that such benefits are unique to CLMs, whereas methods like CFM cannot natively model multiple trajectories. While multi-trajectory training and inference are related to integration over function spaces, exploring this connection is beyond the scope of our work and we leave it for future exploration.

# 5 EXPERIMENTS

We evaluate CaLMFlow on synthetic datasets to showcase the advantages of integral equations for modeling high-dimensional dynamical systems, and apply it to single-cell data generation, demonstrating its ability to model complex distributions and leverage natural language understanding.

## 5.1 SYNTHETIC DATASETS

### 5.1.1 HIGH DIMENSIONAL DATA

Stiffness is a well-known challenge in the numerical integration of ODEs (Kushnir & Rokhlin, 2012; Zappala et al., 2024), particularly in systems with high dimensionality. Conversely, the embedding dimensions of causal language models (CLMs) are inherently large, as demonstrated by pretrained models like GPT-2, which has an embedding dimension of 768. As such, we hypothesize that CaLMFlow is better at modeling high dimensional data than ODE-based methods. To evaluate

| Method ↓ Metric → | Gaussian → 2 Gaussians | | Gaussian → 8 Gaussians | | Gaussian → 2 Moons | |
|---|---|---|---|---|---|---|
| | 2-Wass (↓) | MMD (↓) | 2-Wass (↓) | MMD (↓) | 2-Wass (↓) | MMD (↓) |
| Data Dimension = 100 | | | | | | |
| CFM | $5.483 \pm 0.569$ | $0.324 \pm 0.001$ | $4.846 \pm 0.054$ | $0.224 \pm 0.005$ | $5.061 \pm 0.103$ | $0.194 \pm 0.004$ |
| CFM-OT | $5.494 \pm 0.517$ | $0.322 \pm 0.001$ | $4.795 \pm 0.031$ | $0.227 \pm 0.004$ | $5.013 \pm 0.058$ | $0.195 \pm 0.004$ |
| CFM-SB | $5.504 \pm 0.446$ | $0.336 \pm 0.001$ | $4.914 \pm 0.038$ | $0.236 \pm 0.005$ | $5.294 \pm 0.042$ | $0.218 \pm 0.004$ |
| CaLMFlow | $\mathbf{3.137 \pm 1.028}$ | $\mathbf{0.211 \pm 0.003}$ | $\mathbf{2.317 \pm 0.226}$ | $\mathbf{0.014 \pm 0.001}$ | $\mathbf{2.944 \pm 0.195}$ | $\mathbf{0.018 \pm 0.002}$ |
| Data Dimension = 1000 | | | | | | |
| CFM | $25.064 \pm 1.291$ | $0.488 \pm 0.001$ | $23.294 \pm 0.166$ | $0.402 \pm 0.004$ | $23.428 \pm 0.187$ | $0.363 \pm 0.002$ |
| CFM-OT | $25.131 \pm 1.209$ | $0.490 \pm 0.001$ | $23.116 \pm 0.118$ | $0.407 \pm 0.003$ | $23.339 \pm 0.133$ | $0.363 \pm 0.003$ |
| CFM-SB | $25.053 \pm 1.558$ | $0.493 \pm 0.001$ | $23.211 \pm 0.078$ | $0.412 \pm 0.003$ | $23.805 \pm 0.132$ | $0.373 \pm 0.004$ |
| CaLMFlow | $\mathbf{11.027 \pm 3.853}$ | $\mathbf{0.261 \pm 0.003}$ | $\mathbf{8.272 \pm 0.272}$ | $\mathbf{0.073 \pm 0.001}$ | $\mathbf{13.423 \pm 0.258}$ | $\mathbf{0.039 \pm 0.001}$ |

Table 1: Performance comparison of CaLMFlow and CFM variants across different distribution pairs (Gaussian → 2 Gaussians, Gaussian → 8 Gaussians, Gaussian → 2 Moons) and dimensions (100, 1000). We report 2-Wasserstein distance (2-Wass) and Maximum Mean Discrepancy (MMD) ($\mu \pm \sigma$ over 5 runs), with best results highlighted in bold. As the dimensionality increases, CaLMFlow consistently outperforms CFM variants, achieving lower 2-Wasserstein distances and MMD values, particularly in high-dimensional settings where we expect traditional ODE-based methods, such as CFM, struggle.

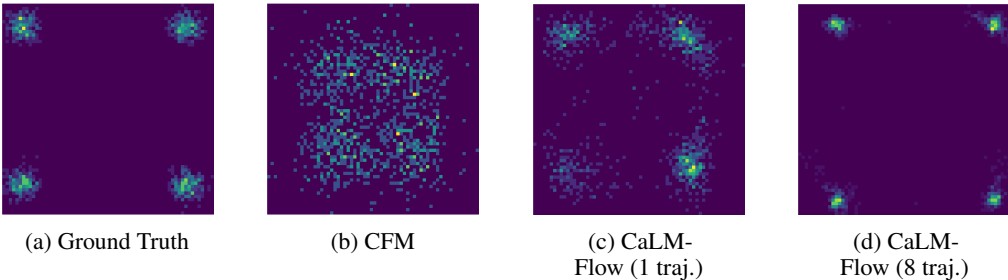

(a) Ground Truth     (b) CFM     (c) CaLM-Flow (1 traj.)     (d) CaLM-Flow (8 traj.)

Figure 2: Heatmaps of the ground truth 4 Gaussians dataset (2a) and that generated by CFM (2b), CaLMFlow (1 traj.) (2c), and CaLMFlow (8 traj.) (2d). Both variants of CaLMFlow generate a distribution that closely matches the ground truth, with the 8-trajectory version further enhancing performance by distributing the data more evenly and accurately.

the robustness of CaLMFlow in high-dimensional settings, we compare its performance against traditional ODE-based methods, which typically degrade as dimensionality increases.

Our results, summarized in Table 1, demonstrate that while CFM breaks down at higher dimensions, CaLMFlow maintains strong performance. This suggests that CaLMFlow is an effective alternative to ODE-based approaches for modeling high-dimensional problems, providing stability and accuracy where methods like CFM fail.

### 5.1.2 MULTI-TRAJECTORY CONTEXT

While approaches like CFM focus on modeling the flow of individual points, CaLMFlow is able to sample multiple trajectories and model them simultaneously. The results in Table 2 show this approach improves the performance of CaLMFlow on generating synthetic data. Visualizations of generated results, as shown in Figure 2, further demonstrate that modeling several trajectories at the same time allows CaLMFlow to distribute data more evenly and accurately by leveraging trajectory context.

### 5.2 SINGLE-CELL GENERATION

We apply CaLMFlow to immune tissue single-cell expression data (Dong et al., 2023) to demonstrate its effectiveness in both unconditional and conditional generation of complex, high-dimensional real-world data. We utilize the first 1,000 principal components of the gene expression data as

| Method ↓ Metric → | 2-Wass (↓) | MMD (↓) | 2-Wass (↓) | MMD (↓) |
|---|---|---|---|---|
| | 2 Gaussians → 3 Gaussians in 100 Dimensions | | 2 Gaussians → 4 Gaussians in 100 Dimensions | |
| CFM | $6.4344 \pm 0.2738$ | $0.0293 \pm 0.0008$ | $6.3400 \pm 0.3185$ | $0.0206 \pm 0.0003$ |
| CaLMFlow (1 traj.) | $4.0898 \pm 0.1842$ | $0.02021 \pm 0.0016$ | $5.6608 \pm 0.3272$ | $0.0148 \pm 0.0007$ |
| CaLMFlow (8 traj.) | $\mathbf{2.8363 \pm 0.3868}$ | $\mathbf{0.0149 \pm 0.0016}$ | $\mathbf{3.6119 \pm 0.3213}$ | $\mathbf{0.0058 \pm 0.0005}$ |

Table 2: Comparisons of CFM, CaLMFlow, and CaLMFlow with multiple trajectories on different distribution pairs in 100 dimensions. We report 2-Wasserstein distance (2-Wass) and Maximum Mean Discrepancy (MMD), averaged over 5 runs. Best results are highlighted in bold. The table shows that CaLMFlow, especially when utilizing multiple trajectories, significantly outperforms CFM in terms of both fit and distribution accuracy. This demonstrates the advantage of leveraging multi-trajectory modeling in CaLMFlow.

| Method ↓ Metric → | MMD(↓) | 2-Wasserstein(↓) | Leiden KLD(↓) | adMMD(↓) |
|---|---|---|---|---|
| CFM | $0.0763 \pm 0.0275$ | $0.0158 \pm 0.0043$ | $0.0330 \pm 0.0027\text{e-}2$ | $9.3568\text{e-}4 \pm 0.7058\text{e-}4$ |
| CFM-OT | $0.0893 \pm 0.0193$ | $0.0149 \pm 0.0012$ | $0.0324 \pm 0.0039\text{e-}2$ | $9.1720\text{e-}4 \pm 0.4719\text{e-}4$ |
| CFM-SB | $0.0998 \pm 0.0050$ | $0.0151 \pm 0.0024$ | $0.0338 \pm 0.0045\text{e-}2$ | $9.5234\text{e-}4 \pm 0.3037\text{e-}4$ |
| DDPM | $0.0709 \pm 0.0010$ | $0.0348 \pm 0.0068$ | $0.0364 \pm 0.0101\text{e-}2$ | $3.8040\text{e-}4 \pm 0.1516\text{e-}4$ |
| scVI | $0.1326 \pm 0.0230$ | $0.0349 \pm 0.0020$ | $0.0360 \pm 0.0096\text{e-}2$ | $11.1673\text{e-}4 \pm 0.4967\text{e-}4$ |
| scGPT | $0.3118 \pm 0.0063$ | $0.4716 \pm 0.0741$ | — | $18.1949\text{e-}4 \pm 0.0531\text{e-}4$ |
| CaLMFlow (1 traj.) | $0.0060 \pm 0.0002$ | $0.0100 \pm 0.0006$ | $\mathbf{0.0311 \pm 0.0045\text{e-}2}$ | $2.4795\text{e-}4 \pm 0.0460\text{e-}4$ |
| CaLMFlow (5 traj.) | $\mathbf{0.0031 \pm 0.0001}$ | $\mathbf{0.0087 \pm 0.0006}$ | $0.0331 \pm 0.0158\text{e-}2$ | $\mathbf{1.8039\text{e-}4 \pm 0.0239\text{e-}4}$ |

Table 3: Unconditional single-cell generation results comparing generated data to ground truth data. To evaluate CaLMFlow's ability to accurately generate its training single-cell dataset (Dong et al., 2023), we computed distributional metrics MMD, 2-Wasserstein, KLD, and adMMD (MMD with a $k$-NN-based adaptive kernel) across 5 seeds. Our default CaLMFlow outperforms all methods across all metrics including CFM and its variants CFM-OT and CFM-SB, demonstrating CaLMFlow's ability to model the data distribution. Further improvement is seen with CaLMFlow (5 traj.), showing the benefit of multi-trajectory tokenization. scGPT's Leiden KLD score is omitted due to the model's poor performance on this metric being less informative for comparison purposes. See Figure 6 for a visual comparison of CaLMFlow and CFM. Experimental details are in Appendices A.1.1 and A.1.2 .

features. The dataset comprises annotations for 7 cell types, 10 perturbations, and 2 chronicities, leading to 140 unique combinatorial labels. In the unconditional generation experiment, the model generates the overall target distribution from Gaussian noise as initial conditions, regardless of labels. In the conditional generation experiment, five combinations of the labels are held out as a test set, and the models are tasked with generating the held-out target distribution conditioned on the unseen combinatorial labels.

Our method is benchmarked against several models, including CFM (Tong et al., 2024) and its variants CFM-OT and CFM-SB, the denoising diffusion probabilistic model (DDPM) (Ho et al., 2020), single-cell generative models scVI (Lopez et al., 2018) and scGPT (Cui et al., 2023), and the Compositional Perturbation Autoencoder (Lotfollahi et al., 2023), depending on the task. To assess the quality of data generated by each model, we compute distributional metrics, such as maximum mean discrepancy (MMD), 2-Wasserstein distance, and Kullback-Leibler Divergence (KLD), between model-generated data and the ground truth test data. For KLD , we use Leiden clustering to generate a distribution of points across clusters (see Appendix A.1.1 for details).

### 5.2.1 UNCONDITIONAL GENERATION OF SINGLE-CELL DATA

The results in Table 3, demonstrate that CaLMFlow consistently outperforms CFM and all other methods across all metrics. Furthermore, as illustrated in Figure 6, CaLMFlow generates cells with distributions more closely aligned to the ground truth data compared to other methods. These findings underscore CaLMFlow's superior performance in capturing and reproducing the complex high-dimensional distributions inherent in single-cell expression data.

| Method ↓ Metric → | MMD(↓) | 2-Wasserstein(↓) | Leiden KLD(↓) | Inception Score(↑) | adMMD(↓) |
|---|---|---|---|---|---|
| CFM | $0.1105 \pm 0.0135$ | $0.0435 \pm 0.0046$ | $0.1076 \pm 0.0049$ | $3.1747 \pm 0.0196$ | $0.0045 \pm 0.0003$ |
| CFM-OT | $0.1082 \pm 0.0140$ | $0.0547 \pm 0.0066$ | $0.1223 \pm 0.0060$ | $2.9794 \pm 0.0261$ | $0.0045 \pm 0.0004$ |
| CFM-SB | $0.1118 \pm 0.0031$ | $0.0460 \pm 0.0066$ | $0.1033 \pm 0.0036$ | $3.1477 \pm 0.0207$ | $0.0046 \pm 0.0002$ |
| scVI | $0.1654 \pm 0.0052$ | $0.5609 \pm 0.0427$ | — | $1.0010 \pm 0.0008$ | $0.0059 \pm 0.0000$ |
| scGPT | $0.2003 \pm 0.0074$ | $0.3513 \pm 0.0274$ | — | $1.0000 \pm 0.0000$ | $0.0062 \pm 0.0000$ |
| CPA* | — | $0.2802$ | — | $1.5831$ | $0.0120$ |
| CaLMFlow (R.I.) | $0.0350 \pm 0.0004$ | $0.0187 \pm 0.0006$ | $0.0897 \pm 0.0059$ | $3.6976 \pm 0.0297$ | $0.0026 \pm 0.0001$ |
| CaLMFlow (N.L.) | $\mathbf{0.0181 \pm 0.0005}$ | $\mathbf{0.0150 \pm 0.0002}$ | $\mathbf{0.0202 \pm 0.0016}$ | $\mathbf{3.9603 \pm 0.0391}$ | $\mathbf{0.0020 \pm 0.0000}$ |

Table 4: Single-cell perturbation response prediction comparison in terms of fit ($\mu \pm \sigma$ over five repeated runs). CFM, scVI, scGPT and CPA are compared. Best results in bold. For CaLMFlow, R.I. stands for randomly initialized CLM and N.L. stands for natural language pretrained CLM. "—" in Leiden KLD represents infinite value due to not being able to generate all classes. *: for CPA, no standard deviation is reported as it is deterministic; CPA generated data that resulted in numerical instability when computing MMD, leading to the absence of valid MMD values

| Cell Representation → | Full Cell | | | Top 100 DE Genes | | |
|---|---|---|---|---|---|---|
| Method ↓ Metric → | $R^2$ (↑) | Pearson (↑) | Spearman (↑) | $R^2$(↑) | Pearson(↑) | Spearman(↑) |
| CFM | $0.4138 \pm 0.1916$ | $0.6280 \pm 0.1395$ | $0.6947 \pm 0.0638$ | $0.3928 \pm 0.2481$ | $0.5851 \pm 0.2262$ | $0.5183 \pm 0.2432$ |
| CFM-OT | $0.8431 \pm 0.0247$ | $0.9181 \pm 0.0134$ | $0.8693 \pm 0.0227$ | $0.8315 \pm 0.0676$ | $0.9111 \pm 0.0369$ | $0.8929 \pm 0.0519$ |
| CFM-SB | $0.4541 \pm 0.1747$ | $0.6623 \pm 0.1248$ | $0.7148 \pm 0.0500$ | $0.4182 \pm 0.2423$ | $0.6127 \pm 0.2069$ | $0.5584 \pm 0.2054$ |
| scVI | $0.1070 \pm 0.0092$ | $0.3268 \pm 0.0137$ | $0.0050 \pm 0.0118$ | $0.3534 \pm 0.0668$ | $0.5919 \pm 0.0554$ | $0.2993 \pm 0.2755$ |
| scGPT | $0.2130 \pm 0.0347$ | $0.4598 \pm 0.0398$ | $0.3813 \pm 0.1029$ | $0.3387 \pm 0.0778$ | $0.5785 \pm 0.0640$ | $0.3928 \pm 0.1225$ |
| CPA | $0.5986 \pm 0.0459$ | $0.7731 \pm 0.0300$ | $0.9185 \pm 0.0151$ | $0.8545 \pm 0.0790$ | $0.9234 \pm 0.0432$ | $0.9303 \pm 0.0270$ |
| CaLMFlow (R.I.) | $0.9862 \pm 0.0087$ | $0.9931 \pm 0.0044$ | $0.9422 \pm 0.0245$ | $0.9653 \pm 0.0270$ | $0.9824 \pm 0.0138$ | $0.9721 \pm 0.0213$ |
| CaLMFlow (N.L.) | $\mathbf{0.9887 \pm 0.0076}$ | $\mathbf{0.9943 \pm 0.0038}$ | $\mathbf{0.9468 \pm 0.0192}$ | $\mathbf{0.9762 \pm 0.0130}$ | $\mathbf{0.9880 \pm 0.0066}$ | $\mathbf{0.9803 \pm 0.0149}$ |

Table 5: Single-cell perturbation response prediction comparison in terms of correlation. $R^2$, Pearson $R$ and Spearman $R$ on the full cell and the top 100 most differentially expressed genes are reported ($\mu \pm \sigma$ over five repeated runs and five unique combinatorial labels). For CaLMFlow, R.I. stands for randomly initialized CLM and N.L. stands for natural language pretrained CLM.

### 5.2.2 SINGLE-CELL PERTURBATION RESPONSE PREDICTION

We leverage CLMs' inherent capabilities to encode and comprehend natural language by representing perturbation conditions as simple text prompts (see A.1.3 for details). These prompts are prepended to the embedded flow-matching conditional trajectories and processed through the CLM's tokenizer and embedding layers. For details on conditional encodings for other models, see A.1.3.

Our architecture is based on a small customized Pythia (Biderman et al., 2023) model as the CLM with a comparable number of parameters (see Table 8 in Appendix A.2 for a comparison). To investigate the benefit of natural language capabilities of CLMs, we compare random initialization to natural language pretraining where the weights are copied from a pretrained 160 million parameter Pythia model.

The results shown in Table 4 highlight CaLMFlow's ability to generate data distributions that closely align with the ground truth, outperforming competing models. The correlation statistics shown in Table 5 underscore CaLMFlow's effectiveness in producing realistic cell expression profiles that correspond to the specified combinatorial labels. Notably, both tables show that leveraging pretrained CLM weights enhances CaLMFlow's performance, showcasing the power of utilizing natural language understanding abilities of CLMs in the CaLMFlow framework.

Furthermore, as shown in Figure 3, both variants of CaLMFlow generate data that closely overlaps with the ground truth distribution, demonstrating CaLMFlow's superior ability to model data under unseen conditions. Figure 7 illustrates the distributions produced by each model, showing that CaLMFlow's generated data most accurately reflects the ground truth. In contrast, other models are either unable to differentiate between combinatorial labels or generate unrealistic distributions. These visualizations reinforce the high quality of data generated by CaLMFlow, emphasizing its capability to model complex distributions and effectively utilize natural language prompts.

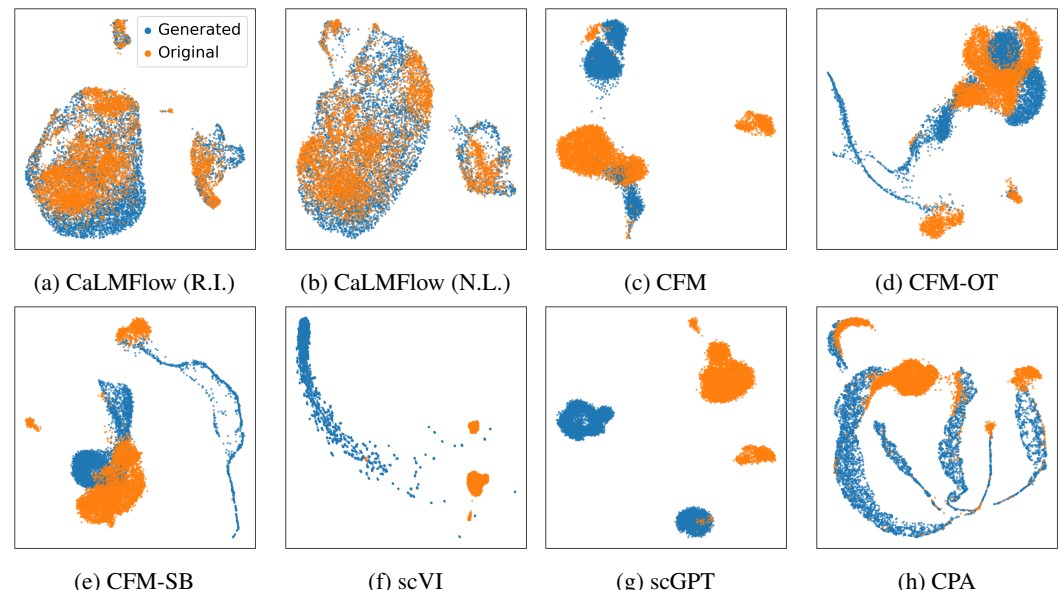

(a) CaLMFlow (R.I.)   (b) CaLMFlow (N.L.)   (c) CFM   (d) CFM-OT

(e) CFM-SB   (f) scVI   (g) scGPT   (h) CPA

Figure 3: Comparison of conditional generation quality across different models for single-cell perturbation data. CaLMFlow (3a and 3b) exhibits strong overlap between generated data distribution (blue) and the ground-truth distribution (orange), highlighting its superior capability to model data with unseen combinatorial perturbations. In contrast, other models struggle to produce realistic samples. For CaLMFlow, R.I. refers to randomly initialized CLM, and N.L. refers to natural language pretrained CLM.

## 6 ABLATION EXPERIMENTS

### 6.1 TEMPERATURE

To investigate the impact of the temperature parameter on CaLMFlow's performance at inference, we varied the temperature for the 8-Gaussians to 2-Moons dataset. Figure 4 shows the best MMD and 2-Wasserstein values at $\tau = 0.2$, where the generated data closely matches the ground truth. Deviations from this value lead to less accurate transformations. Interestingly, it has been empirically found that the optimal temperature in LLMs is often below 1.0 to mitigate inference noise, an observation that aligns with our findings. The experiment also highlights the importance of the VAE component, as its removal significantly degrades performance.

### 6.2 NUMBER OF TIME POINTS

To evaluate the impact of the number of time points on CaLMFlow's performance, we varied the number of time points during training and inference for the 8 Gaussians to 2 Moons dataset. Figure 8 shows that as the number of time points increases, both the MMD and 2-Wasserstein consistently decrease, indicating improved model accuracy. This demonstrates that increasing the number of time points improves CaLMFlow's ability to capture the transformation dynamics, leading to better performance.

### 6.3 NUMBER OF SPATIOTEMPORAL TOKENS AND TRAJECTORIES

To test the impact of spatiotemporal tokenization, we use a similar setup as in Tong et al. (2024) on the MNIST dataset (details in Subsection 4.1). All key hyperparameters, optimizers, and training configurations were kept identical to ensure consistency. Our results, as shown in Table 6, demonstrate that CaLMFlow outperforms other methods and increasing the number of spatial tokens improves inception scores. Similarly, to test the impact of multi-trajectory tokenization, we varied the number of trajectories used to transform data from 2-Moons to 8-Gaussians toy dataset. Table 9 in Appendixe C shows a marked improvement in MMD and 2-Wasserstein with the addition of more trajectories.

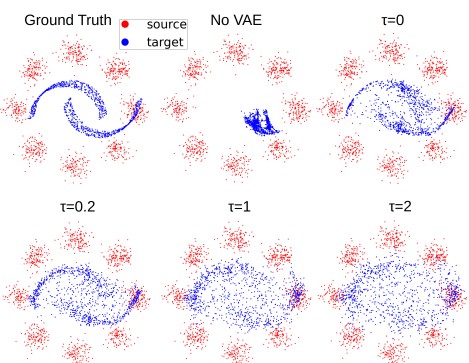 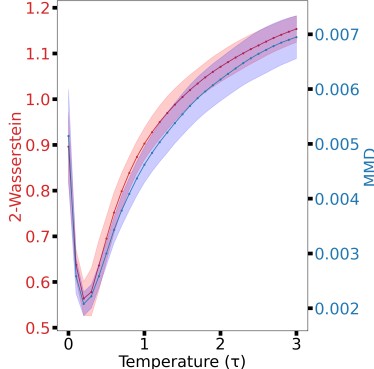

Figure 4: Ablation results on temperature. **Left**: CaLMFlow generated data from 8gaussians to 2moons, using different temperature values. **Right**: 2-Wasserstein and MMD performances as a function of temperature. The plots show that a low, non-zero temperature value ($\tau$=0.2) produces the best performance and that the VAE is necessary.

| Method ↓ | Inception Score (↑) |
|---|---|
| DDPM | 7.1519 ± 0.3456 |
| CFM | 8.9353 ± 0.2334 |
| CFM-OT | 7.5515 ± 0.2935 |
| CaLMFlow (1 Space tokens) | 8.9698 ± 0.1817 |
| CaLMFlow (2 Space tokens) | 8.9619 ± 0.1281 |
| CaLMFlow (4 Space tokens) | 9.1175 ± 0.2103 |
| CaLMFlow (8 Space tokens) | **9.4278 ± 0.1845** |

Table 6: Comparison of inception scores on the MNIST dataset. CaLMFlow (S.T. stands for Space Tokens) outperforms other methods. The results show an improvement in inception scores as the number of space tokens increases, with CaLMFlow (8 Space tokens) achieving the highest score with statistical significance. A comparison of generated images between CFM, CFM-OT, CFM-SB, and CaLMFlow can be found in Figure 9

.

## 7 CONCLUSION AND FUTURE WORK

We introduce CaLMFlow, a novel framework for flow matching that leverages causal language models by casting flow matching as a Volterra integral equation. CaLMFlow outperforms traditional flow matching models like CFM, especially on high-dimensional datasets, such as single-cell generation and perturbation response prediction. It generates more accurate and realistic data in both synthetic and real-world tasks. Future work will formalize CaLMFlow's multi-trajectory approach using integral equations over function spaces and explore its potential as an iterative solver to refine entire trajectory outputs, enhancing its ability to model systems with complex global dynamics.

**Reproducibility Statement** We have supported reproducibility by detailing our experimental implementations, metric computations, and data sources in the appendices. The main text and supplementary materials thoroughly explain our models and methods, along with how the metrics were computed.

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

## A  DETAILED EXPERIMENT SETUP

All models are implemented in PyTorch (Paszke et al., 2019), trained using the Adam optimizer (Kingma & Ba, 2017). Additionally, FlashAttention-2 (Dao, 2024) is employed to accelerate training.

For the synthetic experiments, we utilize GPT-2 (Radford et al., 2019) as the CLM, while in the single-cell experiments, we adopt Pythia (Biderman et al., 2023). Both models are hosted on the open-source LLM platform Hugging Face (Wolf et al., 2020).

### A.1  SINGLE CELL GENERATION

#### A.1.1  DETAILS ON DISTRIBUTION AND CORRELATION METRICS

**MMD and 2-Wasserstein Distance**  Distributional metrics, even with kernel methods, are shown to suffer greatly from curse of dimensionality as shown in previous work(Reddi et al., 2014; Arias-Castro et al., 2018). Therefore, rather than computing MMD and 2-Wasserstein in the original 1000-dimensional feature space, we first embed the ground truth and generated distribution together with UMAP and compute the metrics in the joint UMAP space. The dimension of the UMAP is chosen to be 10.

For MMD, we adopt a radial basis function (RBF) kernel $k : \mathcal{X} \times \mathcal{X} \to \mathbb{R}$ with

$$k(x, y) = \exp\left(-\|x - y\|^2\right)$$

The estimator of MMD between two the generated distribution $P = (x_1, ..., x_N)$ and the ground truth distribution $Q = (y_1, ..., y_N)$ is computed as

$$\text{MMD}(P, Q) = \frac{1}{N(N-1)} \sum_{i \neq j} k(x_i, x_j) + \frac{1}{N(N-1)} \sum_{i \neq j} k(y_i, y_j) - \frac{2}{N^2} \sum_{i=1}^{N} \sum_{j=1}^{N} k(x_i, y_j)$$

The 2-Wasserstein distance is computed as

$$W_2(P, Q) = \inf_{\pi} \sqrt{\left(\frac{1}{n} \sum_{i=1}^{n} \|x_i - y_{\pi(i)}\|^2\right)},$$

where $\pi$ are permutations on $n$ elements. This is implemented with the Python Optimal Transport toolbox (Flamary et al., 2021).

We compute adaptive MMD scores using an adaptive Radial Basis Function (RBF) kernel, where the bandwidth is dynamically adjusted according to the local density of points.

$$K(X, Y) = \exp\left(-\frac{\|X - Y\|^2}{2\sigma_X \sigma_Y}\right)$$

The bandwidth for each point is determined based on the distances to its $k$-nearest neighbors (k-NN), enabling the kernel to adapt to varying local densities. This adaptive mechanism enhances the kernel's ability to capture distributional differences in non-homogeneous data, improving the sensitivity of the MMD to local structure.

**Leiden KL Divergence**    We employ the Leiden clustering algorithm (Traag et al., 2019) to identify community structures within the ground truth data. We run Leiden clustering with the resolution parameter set to 0.3, resulting in 8 clusters as shown in Figure 5a. We then train a simple two-layer MLP with ReLU activation to predict the Leiden labels from the ground truth data, achieving a prediction accuracy of 95% on a held-out test set. This MLP classifier is subsequently used to assign predicted Leiden cluster labels to data generated by different models. To assess model performance, we compute the KL divergence between the distributions of Leiden clusters in the ground truth and generated data.

**Inception Score**    For the conditional generation task, to assess the quality of the generated cells, we utilize the Leiden cluster labels and the trained MLP classifier to compute an inception score, inspired by its use in the computer vision domain (Salimans et al., 2016). Let $p(y|\mathbf{x})$ represent the conditional distribution of the MLP classifier assigning a label $y$ to a sample $\mathbf{x}$, the Inception Score is calculated as

$$\text{IS} = \exp\left(\mathbb{E}_x \text{KL}(p(y|\mathbf{x})||p(y)\right),$$

where $p(y)$ is the marginal distribution of the label $y$.

**Correlation Metrics**    For the conditional generation task, we additionally compute correlation metrics such as $R^2$, Pearson $R$, and Spearman $R$ between the average generated cells and the average ground truth cells for each condition, to assess how well the models generate cells based on the specified conditions. These metrics are calculated per combinatorial label and then averaged. For each condition (cell type, perturbation, and chronicity) we identify the top 100 most differentially expressed genes by comparing them to control cells of the same type and chronicity but without stimulation.

### A.1.2    SINGLE CELL UNCONDITIONAL GENERATION MODEL IMPLEMENTATIONS

CaLMFlow's architecture is a small Pythia model with 64 hidden dimensions, a 256-dimensional feedforward layer, 4 attention heads, and 2 blocks. The encoder for the cell expression vector, latent mean, latent sigma and the decoder were all 2-layer MLPs, with input and output dimensions adjusted for the number of spatial tokens. When using pretrained weights, we take the upper left block of all weight matrices of the same dimension as our customized model. We used the standard setup for CFM following the torchCFM library—a 4-layer MLP with intermediate width 1024 for CFM, CFM-OT, and CFM-SB. At inference, we use 100 function evaluations using the adaptive "dopri5" solver in the NeuralODE package with the default parameters. For DDPM, we use a 2-layer MLP with 2048 intermediate dimension, and 100 denoising steps. We found similar performance with deeper and wider layers, so we used a smaller model to account for the number of parameters. For scVI, we trained on the full gene expression since its zero-inflated negative binomial decoder is not meant to handle non-sparse data. At inference, we randomly sample from the prior distribution to generation samples with the decoder. For scGPT, we trained the model with a 5000 gene context using the masked decoding decoding scheme from the official repository with masking ratios 0.25, 0.5, and 0.75. At inference, we mimicked the training procedure as closely as possible by using 2500 context genes and masking 2500 genes until an entire cell is generated. The first 2500 genes are taken from a randomly sampled ground truth cell expression vector.

The below table includes a comparison of each model's number of trainable parameters.

### A.1.3    SINGLE CELL CONDITIONAL ENCODING

**CaLMFlow**    We use the following templates to form the natural language prompts:

For perturbed cells, the prompts are:

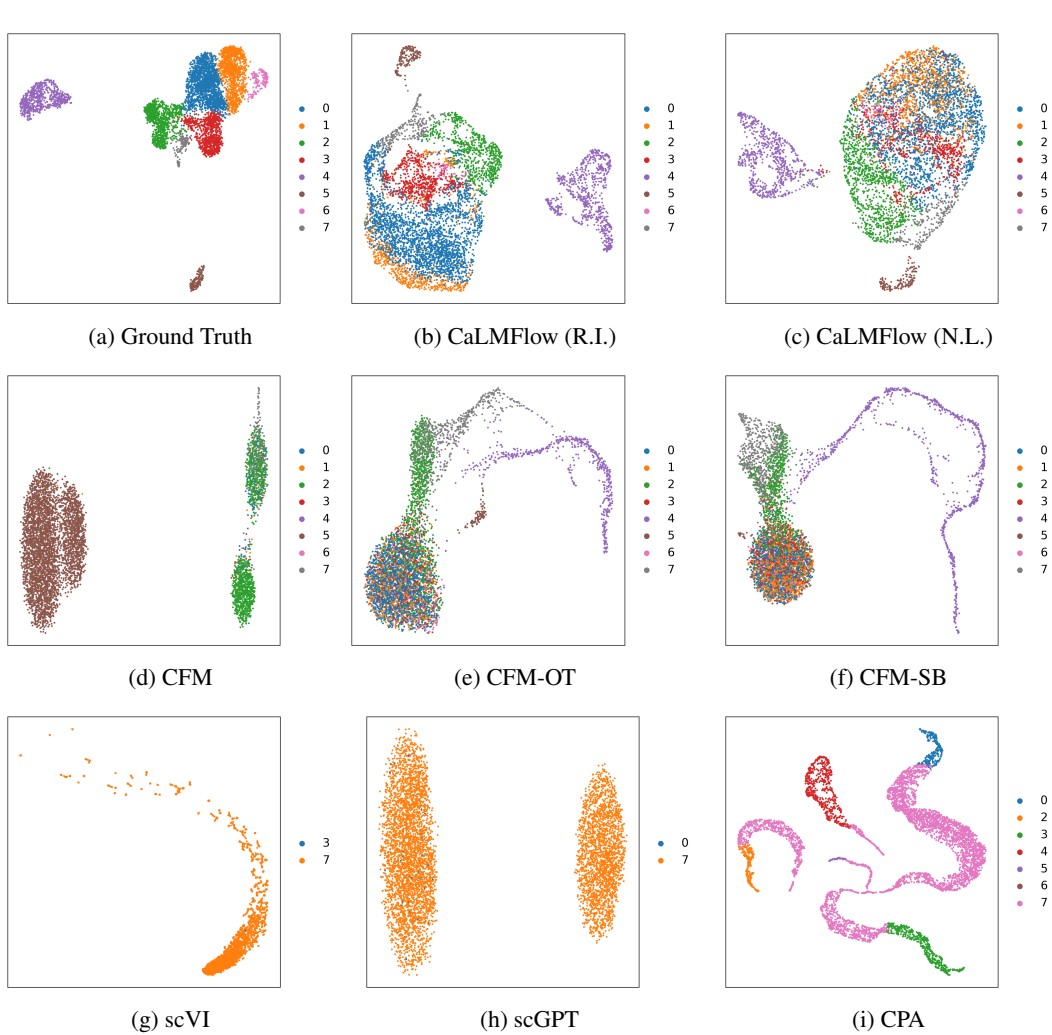

Figure 5: Comparison of the ground truth data and model-generated data, colored by Leiden labels. For the generated data, the Leiden labels are predicted by an MLP classifier trained on the ground truth. Both variants of CaLMFlow successfully generate data spanning all clusters and closely align with the ground truth distribution. In contrast, while CFM, CFM-OT, and CFM-SB generate data across all classes, they fail to differentiate between them, indicating a mismatch in the underlying community structures. Models such as scVI, scGPT, and CPA are unable to generate data for some classes altogether.

| Model | Number of trainable parameters |
|---|---|
| CFM (all variants) | 4,150,248 |
| DDPM | 12,696,552 |
| scVI | 14,018,766 |
| scGPT | 51,330,049 |
| CaLMFLow (custom Pythia-160M) | 2,194,104 |

Table 7: Comparison of the number of parameters of each model

```
"Generate a {cell type} cell stimulated with {perturbation} and
exposure {chronicity}:"
```

For control cells, the prompts are:

```
"Generate a {cell type} cell with no stimulation and exposure
{chronicity}:"
```

For example, one sample prompt for a perturbed cell is:

```
"Generate a CD4 T cell stimulated with IL-6 and exposure acute:"
```

and one sample prompt for a control cell is:

```
"Generate a B cell with no stimulation and exposure chronic:"
```

The prompts are tokenized using the Hugging Face tokenizer class `AutoTokenizer.from_-pretrained("EleutherAI/pythia-160m")` and embedded using the embedding layers of a customized Pythia-160M model. The embedding vectors are prepended to the corresponding conditional flow matching trajectories and processed by the CLM. Since we don't train the CLM to generate text responses, no losses are applied on the text prompts during training.

During inference, we sample combinatorial labels from the test dataset, form prompts as above, sample initial Gaussian noises, tokenize and prompt the CLM to generate the trajectories and consequently the target distribution.

**CFM**  Conditional Flow Matching doesn't provide an option for conditional generation out of the box, so adapted the torchCFM python library. In order to allow CFM to generate contionally, we create 3 one-hot vectors corresponding to each of the 3 conditions: cell type, perturbation, and chronicity. With 7 cell types, 10 perturbations, and 2 exposures, we obtain a 19-dimensional vector once concatenated together. This conditioning vector is then appended to the data vector, and CFM learns the vector field as usual. The NeuralODE python package was adapted to handle conditioning vectors so we could apply the same adaptive solver used in Tong et al. (2024).

**scVI**  To condition scVI, we applied the same one-hot encoding scheme as CFM to the latent vectors prior to decoding the posterior parameters.

**scGPT**  We trained scGPT (Cui et al., 2023) with a 5000 gene context using the masked decoding decoding scheme from the official repository with masking ratios 0.25, 0.5, and 0.75. The conditioning vectors were one-hot encoded, and we used learned embedding matrices for each type of condition and added the learned embedding to the cell expression. At inference, we applied the same procedure, always using 2500 context genes and masking 2500 genes until an entire cell is generated. The first 2500 genes are generated using a sampled ground truth cell expression vector from the test dataset, giving an advantage to scGPT other approaches did not have. Despite this, scGPT still showed poor performance relative to other methods.

**CPA**  CPA is a framework of perturbation response prediction. We follow their setup as in their official GitHub repository by setting our perturbation as `perturbation_key` and `cell type` and `chronicity` as categorical covariates.

## A.2 Multi-trajectory Experiments, Ablations on Temperature and Number of Time Points

**CaLMFlow** The model architecture for CaLMFlow consists of a simple MLP to embed the input into the encoder, represented by GPT-2. The dimensionality of the embeddings and hidden states in GPT-2 is set to 32. The GPT-2 Transformer encoder uses 4 hidden layers with 1 attention head per attention layer. The decoder is implemented as a VAE, with a latent dimension of 16 and a hidden dimension of 32. For CaLMFlow, the scaling hyperparameter $\beta$, the number of time points, and the temperature parameter $\tau$ are set to empirically optimal values for each task.

**CFM** For CFM, we use the standard architecture described in Tong et al. (2024) for the 2D experiments. Trajectories are evaluated using the *dopri5* option of the adaptive solver.

The below table includes a comparison of each model's number of trainable parameters for the conditional generation task.

| Model | Number of trainable parameters |
|---|---|
| CFM (all variants) | 4,169,704 |
| scVI | 28,033,895 |
| scGPT | 51,342,849 |
| CaLMFLow* (custom Pythia-160M) | 2,606,904 |

Table 8: Comparison of the number of parameters of each model. *For CaLMFlow, the number of parameters excludes the embedding parameters for text, which is approximately 6.5 million parameters.

## B Training and Inference Algorithms

---
**Algorithm 1** Inference Algorithm

---
**Sizes:** Data dimension $D$, hidden dimension $H$, discretize $[0,1]$ into $N$ time steps, $K$ spatial tokens per time step, $M$ trajectories
**Model components:** Embedding function $E_\theta : \mathbb{R}^D \rightarrow \mathbb{R}^H$, projection $S_\theta : \mathbb{R}^H \rightarrow \mathbb{R}^{H \cdot K}$, autoregressive transformer $T_\theta$, mean and variance projections $\mu_\theta, \sigma_\theta : \mathbb{R}^{H \cdot K} \rightarrow \mathbb{R}^H$, decoder $\text{Dec}_\theta$
**Input:** $\mathbf{X} \sim \mathcal{N}(\mathbf{0}, \mathbf{I})$, $\mathbf{X} \in \mathbb{R}^{M \times D}$ (Optional: embedded text sequence of length $P$, $\mathbf{Z}_T \in \mathbb{R}^{P \times H}$)
**for** $n = 1, \ldots, N - 1$ **do**
    $\mathbf{Z} \leftarrow E_\theta(\mathbf{X})$                                     ▷ Embed data into latent space.
    $\mathbf{Z} \leftarrow S_\theta(\mathbf{Z})$           ▷ Expand latent dimension with projection $S$ for spatial tokenization.
    $\mathbf{Z} \leftarrow \text{Reshape}(\mathbf{Z})$    ▷ Input dimension $n \times M \times H \cdot K$, output dimension $n \times K \times M \times H$
    $\mathbf{Z} \leftarrow \text{Flatten}(\mathbf{Z})$     ▷ $\mathbf{Z} = [\mathbf{z}_{111}, \ldots, \mathbf{z}_{1K1}, \ldots, \mathbf{z}_{1KM}, \ldots, \mathbf{z}_{N11}, \ldots, \mathbf{z}_{nKM}] \in \mathbb{R}^{n \cdot K \cdot M \times H}$.
    **if** text **then**
        $\mathbf{Z} \leftarrow \text{Concat}[\mathbf{Z}_T, \mathbf{Z}]$     ▷ Prepend embedded text tokens to embedded continuous data.
    **end if**
    $\mathbf{Z} \leftarrow T_\theta(\mathbf{Z})$
    $\mathbf{Z} \leftarrow \text{Reshape}(\mathbf{Z})$                              ▷ Reshape back to $n \times M \times H \cdot K$.
    $\mathbf{Z}_\mu, \mathbf{Z}_\sigma \leftarrow \mu_\theta(\mathbf{Z}), \mu_\sigma(\mathbf{Z})$
    $\mathbf{Z} \leftarrow \mathbf{Z} \sim \mathcal{N}(\mathbf{Z}_\mu, \mathbf{Z}_\sigma)$
    $\mathbf{X} \leftarrow \text{Concat}[\mathbf{X}, \text{Dec}_\theta(\mathbf{Z}[n])]$
**end for**
**Return:** $\mathbf{X}$

---

**Algorithm 2** Training Algorithm

**Training data:** $\mathcal{D}$
**Sizes:** Data dimension $D$, hidden dimension $H$, discretize $[0,1]$ into $N$ evenly spaced time steps $0 = t_0, t_1, t_2, \ldots, t_N = 1$, $K$ spatial tokens per time step, $M$ trajectories
**Model components:** Embedding function $E_\theta : \mathbb{R}^D \to \mathbb{R}^H$, projection $S_\theta : \mathbb{R}^H \to \mathbb{R}^{H \cdot K}$, autoregressive transformer $T_\theta$, mean and variance projections $\mu_\theta, \sigma_\theta : \mathbb{R}^{H \cdot K} \to \mathbb{R}^H$, decoder $\text{Dec}_\theta$ (Optional: text embedding layer $C_\theta$)
**while** not converged **do**                                              ▷ For stochastic gradient descent
    $\mathbf{X} \sim \mathcal{N}(\mathbf{0}, \mathbf{I})$, $\mathbf{X} \in \mathbb{R}^{M \times D}$
    $\mathbf{Y} \sim \mathcal{D}$, $\mathbf{Y} \in \mathbb{R}^{M \times D}$
    $\mathbf{W} \leftarrow \mathbf{0} \in \mathbb{R}^{N \times M \times D}$
    **for** i=0, 1, $\ldots$, N-1 **do**
        $\mathbf{W}[i] \leftarrow (1 - t_i)\mathbf{X} + t_i \mathbf{Y}$
    **end for**
    $\mathbf{Z} \leftarrow E_\theta(\mathbf{W})$                                       ▷ Embed data into latent space.
    $\mathbf{Z} \leftarrow S_\theta(\mathbf{Z})$               ▷ Expand latent dimension with projection $S$ for spatial tokenization.
    $\mathbf{Z} \leftarrow \text{Reshape}(\mathbf{Z})$    ▷ Input dimension $n \times M \times H \cdot K$, output dimension $n \times K \times M \times H$
    $\mathbf{Z} \leftarrow \text{Flatten}(\mathbf{Z})$     ▷ $\mathbf{Z} = [\mathbf{z}_{111}, \ldots, \mathbf{z}_{1K1}, \ldots, \mathbf{z}_{1KM}, \ldots, \mathbf{z}_{N11}, \ldots, \mathbf{z}_{nKM}] \in \mathbb{R}^{n \cdot K \cdot M \times H}$.
    **if** text **then**
        $\mathbf{Q} \leftarrow$ tokenized text
        $\mathbf{Z}_T \leftarrow C_\theta(\mathbf{Q})$
        $\mathbf{Z} \leftarrow \text{Concat}[\mathbf{Z}_T, \mathbf{Z}]$       ▷ Prepend embedded text tokens to embedded continuous data.
    **end if**
    $\mathbf{Z} \leftarrow T_\theta(\mathbf{Z})$
    $\mathbf{Z} \leftarrow \text{Reshape}(\mathbf{Z})$                                    ▷ Reshape back to $n \times M \times H \cdot K$.
    $\mathbf{Z}_\mu, \mathbf{Z}_\sigma \leftarrow \mu_\theta(\mathbf{Z}), \mu_\sigma(\mathbf{Z})$
    $\mathbf{Z} \leftarrow \mathbf{Z} \sim \mathcal{N}(\mathbf{Z}_\mu, \mathbf{Z}_\sigma)$
    $\mathbf{W}_{\text{pred}} \leftarrow \text{Dec}_\theta(\mathbf{Z})$
    Loss $\leftarrow L(\mathbf{W}[1:], \mathbf{W}_{\text{pred}}[:-1])$                ▷ Loss function define in Equation 5
    Update(Loss)                                        ▷ Standard backpropagation with optimizer.
**end while**

## C    ADDITIONAL RESULTS

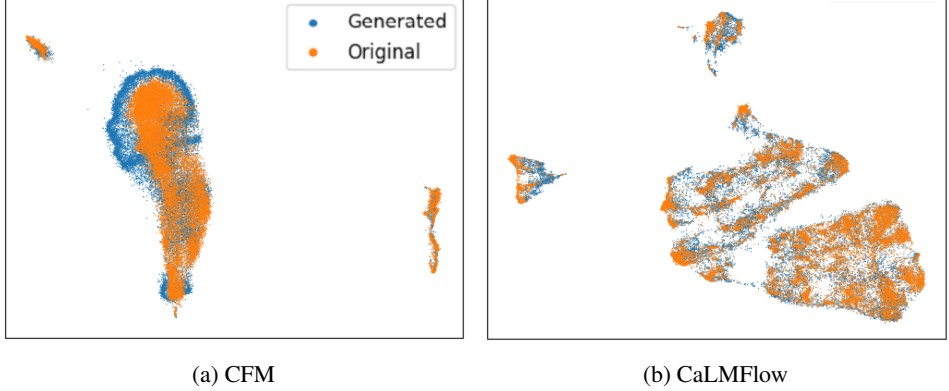

(a) CFM                                    (b) CaLMFlow

Figure 6: Comparison of UMAPs of generated versus ground truth single-cell data between CFM and CaLMFlow. For each model, we generated 20,000 cells and plotted a UMAP with 20,000 randomly sampled cells from the immune cell dataset from Dong et al. (2023). The plots demonstrate that CaLMFlow produces a single-cell distribution more accurately reflecting the ground truth data, in contrast to CFM.

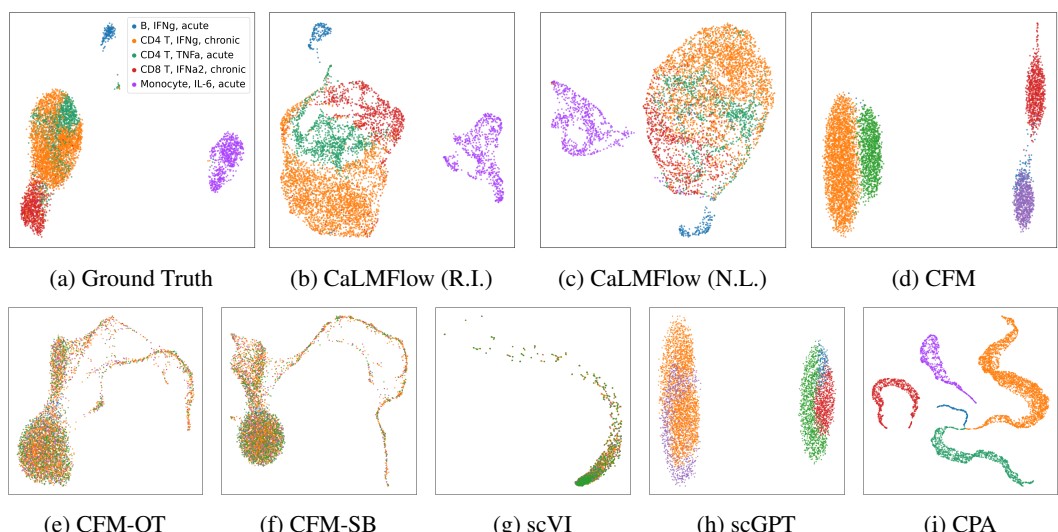

Figure 7: Comparison of conditional generation quality across different models for single-cell perturbation data. CaLMFlow (7b and (7c) generates data that accurately reflects the ground truth distribution across all combinatorial labels (cell type, perturbation, and chronicity), demonstrating its superior ability to understand complex conditions while maintaining a realistic overall data distribution. In contrast, methods like CFM-OT (7e), CFM-SB (7f), and scVI (7g) generate data where labels are blended, indicating an inability to comprehend conditions. While CFM (7d) and scGPT (7h) produce well-separated data, it's easily observed that this does not realistically represent the actual distribution when compared to the ground truth. For CaLMFlow, R.I. refers to randomly initialized CLM, and N.L. refers to natural language pretrained CLM.

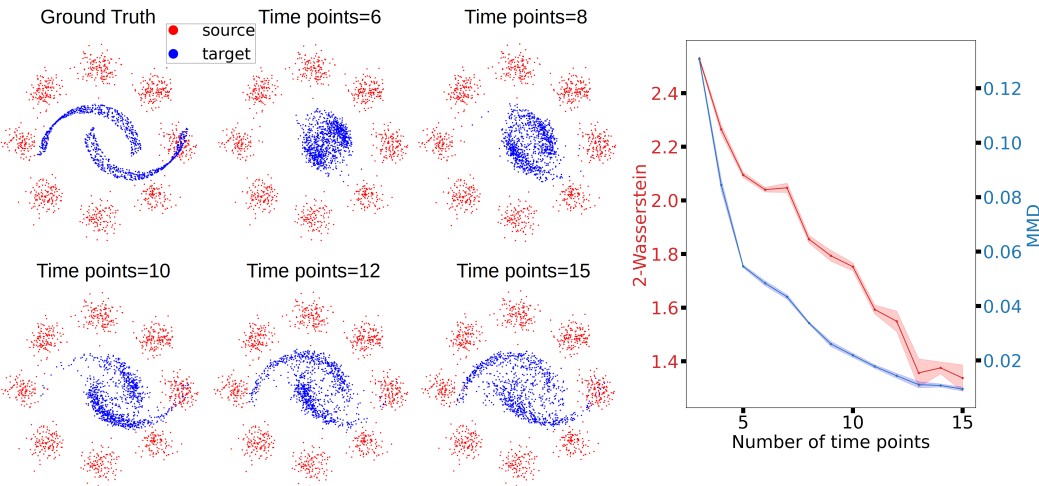

Figure 8: Ablation results on number of time points. **Left**: CaLMFlow generated data from 8gaussians to 2moons, using different number of time points. **Right**: 2-Wasserstein and MMD performances as a function of number of time points. The plots show that increasing the number of time points improves the model performance.

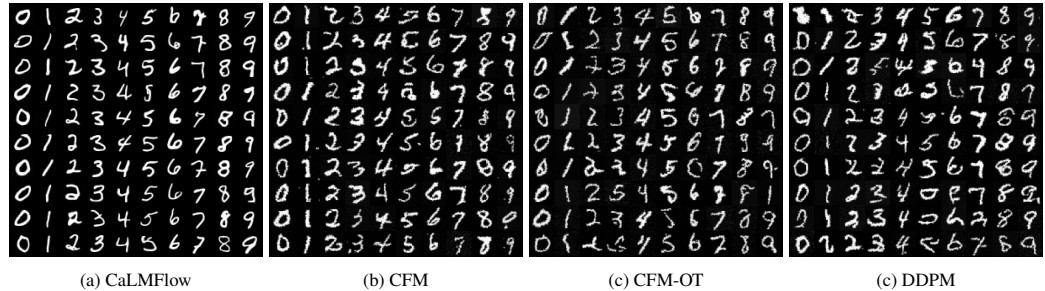

| (a) CaLMFlow | (b) CFM | (c) CFM-OT | (c) DDPM |

Figure 9: Uncurated conditional image generation comparison between CaLMFlow, CFM models, and DDPM on MNIST. CaLMFlow generates higher image quality images compared to both CFM variants and DDPM.

| Method ↓ Metric → | 2-Wass (↓) | MMD (↓) |
|---|---|---|
| CaLMFlow (1 traj.) | $1.3756 \pm 0.0668$ | $0.0057 \pm 0.0002$ |
| CaLMFlow (4 traj.) | $1.0477 \pm 0.0771$ | $0.0041 \pm 0.0003$ |
| CaLMFlow (10 traj.) | $\mathbf{0.8753 \pm 0.1243}$ | $\mathbf{0.0020 \pm 0.0003}$ |

Table 9: Ablation results on multi-trajectory implementations of CaLMFlow transforming 2 moons into 8 Gaussians in 2 dimensions. We evaluate the MMD and 2-Wasserstein between the generated target distribution and ground truth distribution over 5 seeds. The results show increasing the number of trajectories improves CaLMFlow's ability to accurately generate the target distribution.

# D  BANACH SPACES, VOLTERRA INTEGRAL EQUATIONS AND CLMS

A Volterra integral equation of the second kind is defined as:

$$z(t) = z(0) + \int_0^t G(z(s), t, s)ds \tag{9}$$

where $G(z(s), t, s)$ is a Urysohn kernel function encoding the influence of past states on the current state.

We adopt the standard next-token prediction paradigm used in CLMs. Given tokens $x_0$ to $x_k$, the model predicts $x_{k+1}$, where the sequence $(x_0, \ldots, x_k)$ corresponds to portions of the conditional flow matching trajectory. In this section, we give the implementation details of the theoretical discussion given in Section 3.2.

Our training procedure enables the causal language model (CLM) to learn system dynamics by modeling sequences of varying lengths. During training, the model predicts the next state in the trajectory given previous states, similar to next-token prediction in language models, but with tokens representing continuous trajectory states.

Solving a nonlinear integral equation generally requires some iterative procedure, where an initial guess is refined iteratively. Since the evaluation of $z$ at time $t$ requires evaluation of $z$ at all time points between 0 and $t$ due to the integral $\int_0^t G(z_s, t, s)ds$. Observe that once an approximation (guess) $z^j(t)$ has been obtained, all evaluations in parallel for each iteration, since we can integrate using $z^j(t)$, see Zappala et al. (2024) for details. We present the model with sequences starting from the initial state $z_0$ and extending to various lengths (e.g., predicting from $z_0$ to $z_1$, $z_0$ to $z_2$, up to $z_0$ to $z_N$). This trains the model on multiple sub-trajectories, which can be used in inference in an iterative manner to output the complete trajectory.

This training methodology can be viewed through the lens of functional analysis, where we consider the CLM as learning operators on a direct sum of Banach spaces. Each Banach space corresponds to sequences (or discretized trajectories) of a particular length, and the direct sum represents the combination of these spaces. The norm on this direct sum space is defined as the sum of the norms of the individual spaces. By minimizing the loss, which aggregates the errors across all sequence lengths, the model learns to operate effectively on each of these spaces.

Let $\mathcal{B}_k$ denote some Banach space of functions with domain $[0, t_k]$, equipped with an appropriate norm $\| \cdot \|_{\mathcal{B}_k}$, *e.g.*, the $L^2$ space. The direct sum $\mathcal{B} = \bigoplus_{k=1}^N \mathcal{B}_k$ is the direct sum of the $\mathcal{B}_k$ spaces. The total loss $\mathcal{L}$ in this space is the sum of the component norms:

$$\mathcal{L} = \sum_{k=1}^N \| z^i(t) - \hat{z}^i(t) \|_{\mathcal{B}_k} \tag{10}$$

where $z^i$ indicates the ground-truth between $[0, t_i]$, and $\hat{z}^i$ is the model's prediction on $[0, t_i]$. This loss function aggregates the errors over all sequence lengths, reflecting the norm on the direct sum space.

During inference, the model sequentially generates the trajectory by predicting the next state given the current sequence, starting from the initial state $z_0$. At each step, the model uses the operators it has learned during training to map the current sequence to the next state. This process can be interpreted as the model applying learned operators on the respective Banach spaces to generate the trajectory. We note that this process in inference is sequential, but during training is performed in parallel.

The LLM computes:

$$z_{i+1} = f_\theta(z_i, t_i) + \Delta t \sum_{j=0}^i G_\theta(z_j, t_i, t_j),$$

and as $\Delta t \to 0$ we obtain Equation 3, where we note that during inference there is no need of distinguishing between the $z$ used in the integral operator and the prediciton, as the prediction is used iteratively to produce the next $z^{i+1}$, contrary to training where the ground truth is used to perform the process in parallel.

By framing the problem as a sequence modeling task, the LLM effectively approximates the solution to the integral equation. The sequence of states $\{z_t\}$ can be viewed as tokens in a sequence, where each state depends on all previous states due to the integral over past times $s$ in Equation 3. The LLM, with its inherent capability to model long-range dependencies through attention mechanisms, captures this dependence without the need to explicitly compute $f_\theta$ and $G_\theta$.

In practice, we model Equation 3 using the LLM as follows:

- The LLM models the conditional distribution of each $z_{t_{i+1}}$ given the past states $\{z_{t_0}, \ldots, z_{t_i}\}$:

$$p(z_{t_{i+1}} | z_{t_0}, z_{t_1}, \ldots, z_{t_i}) = \text{LLM}(z_{t_0}, z_{t_1}, \ldots, z_{t_i}), \tag{11}$$

  which implements an integral depending on $z_{t_0}, \ldots, z_{t_i}$, i.e. $z(s)$ with $s \in [0, t_i]$.

We do not define an iterative procedure here for the training, but leverage the use of LLMs and learn to predict $n$ functions $z^i$, each of which model the solution $z(t)$ (i.e. the ground truth) between $t = 0$ and $t = t_i$. This is in line with the use of LLMs and allows us to formulate the solver procedure on a direct sum Banach space, $X = \bigoplus_{i=1}^n L^2([0, t_i])$, where the target of each $L^2([0, t_i])$ is the ground truth of the trajectory. In inference, we apply an iterative procedure where the model predicts each component $z^i$, and uses this to approximate the integral to compute $z^{i+1}$:

$$z_t^{i+1} = f(z_t, t) + \int_0^{t_i} G(z_s^i, t, s) ds, \tag{12}$$

for $t \in [t_i, t_{i+1}]$, where the integral operator is well defined because $z^i$ is defined over $[0, t_i]$ – i.e. it is an element of $L^2([0, t_i])$ as stated above.

In order to perform an approximate integral over the discretized trajectory, we also need a discretized version of Equation (12), which becomes:

$$z_{i+1} = f_\theta(z_i, t_i) + \sum_{j=0}^i \Delta t_{i+1} G_\theta(z_j, t_{i+1}, t_j), \tag{13}$$

where $\Delta t = t_{i+1} - t_i$, and $z_i = z(t_i)$. Once again, observe that we need each $z_j$ with $j = 0, \ldots, i$, and we use the ground-truth data for this.

We can easily extend the VIE to include both spatial and temporal domains:

$$z(x, t) = f(z, x, t) + \int_0^t \int_\Omega G(z, x, x', t, s) dx' ds. \tag{14}$$

# E  VOLTERRA FLOW MATCHING OBJECTIVE

The goal of this section is to connect the conditional Volterra flow matching object to the CFM objective. We recall some notation from Lipman et al. (2022). Let $t \in [0, 1]$, $u_t(x)$ the marginal time dependent vector field associated with the flow $\phi$, $u_t(x|x_1)$ the conditional time dependent vector field, $v_t(x)$ the model learning the vector field $u_t$, $q$ the data distribution, and $p_t$ the probability density path. Then we can define the flow matching objective and its conditional variant, $\mathcal{L}_{\text{FM}}(\theta) = \mathbb{E}_{t, p_t(x)} \|v_t(x) - u_t(x)\|^2$ and $\mathcal{L}_{\text{CFM}}(\theta) = \mathbb{E}_{t, q(x_1), p_t(x|x_1)} \|v_t(x) - u_t(x|x_1)\|^2$. The key observation is that the gradients of these two objective functions are equivalent:

**Theorem** ((Lipman et al., 2022)). *Assuming that $p_t(x) > 0$ for all $x \in \mathbb{R}^d$ and $t \in [0, 1]$, then, up to a constant independent of $\theta$, $\mathcal{L}_{CFM}$ and $\mathcal{L}_{FM}$ are equal. Hence, $\nabla_\theta \mathcal{L}_{FM}(\theta) = \nabla_\theta \mathcal{L}_{CFM}(\theta)$.*

The crux of the proof of Theorem E rests in the chain of equalities (see E Appendix A for details):

$$\mathbb{E}_{p_t(x)} \langle v_t(x), u_t(x) \rangle = \int \left\langle v_t(x), \int u_t(x|x_1) p_t(x|x_1) q(x_1) dx_1 \right\rangle p_t(x) dx$$

$$= \int \left\langle v_t(x), \int u_t(x|x_1) p_t(x|x_1) q(x_1) dx_1 \right\rangle dx$$

$$= \int \langle v_t(x), u_t(x_1) \rangle p_t(x) q(x_1) dx_1 dx$$

$$= \mathbb{E}_{q(x_1), p_t(x|x_1)} \langle v_t(x), u_t(x|x_1) \rangle.$$

We can expand $v_t$ in the penultimate line as:

$$\int \langle v_t(x), u_t(x_1) \rangle p_t(x) q(x_1) dx_1 dx = \int \left\langle \int v_t(x|x_1') dx_1', u_t(x_1) \right\rangle p_t(x) q(x_1) dx_1 dx.$$

Note that CaLMFlow takes conditioned inputs during training, i.e. it is history-aware. One approach to training CaLMFlow like CFM is to randomly sample pairs of conditional trajectories—one for the target trajectory and one for the input trajectory to generate the flow—and minimize the mean squared error, either of the next time step or the output of a numerical solver. However, in practice we have found our model performance to benefit from using the the CVFM objective in Equation 6 combined with the KL divergence regularizer from Equation 8. We leave the exploration of alternative optimization procedures to future work.

