# OpenReview forum: "CaLMFlow: Volterra Flow Matching using Causal Language Models"
_ICLR.cc/2025/Conference — Submitted to ICLR 2025_

### Official Review · Reviewer_LGVE · 2024-11-03

**Soundness:** 2
**Presentation:** 3
**Contribution:** 1
**Rating:** 3
**Confidence:** 4

**Summary:**

This paper proposes CALMFLOW, an autoregressive modeling method for continuous data. It tokenizes continuous data into small time segments and uses numerical solutions of ODEs to model the data. Experiments are conducted on synthetic data and real cell-perturbation data to evaluate the approach.

**Strengths:**

The writing is clear and accessible.

The idea is easy to understand.

Experiments are conducted on both synthetic and real data, enhancing the validity of the results.

**Weaknesses:**

Trivial Task: The paper focuses on continuous data modeling, which may not present a sufficiently challenging or novel task. The modeling of Gaussian distributions and the MNIST dataset appears trivial, and many existing multimodal methods can already handle cell data modeling effectively.

Mismatched Motivation and Experiment: Although the paper initially highlights the difficulty of solving ODEs, the experiments focus only on continuous data modeling and do not fully support or address the stated challenges with ODEs.

Overclaim of Novelty: The claimed tokenization method seems indistinguishable from standard window-based segmentation techniques, which questions the novelty of the approach.

**Questions:**

Problem Definition: What is the task setup for the synthetic experiments? How many training and test data samples are used, and what does each training sample look like (e.g., is it a two-dimensional Gaussian distribution)?

Parameter Comparison: There is no parameter comparison between different methods, which makes the evaluation less fair for data-fitting tasks.

---

> ### Author Response · Authors · 2024-11-19
> **Official Response by the Authors [1/2]**
>
> We would like to express our sincere gratitude to the reviewer for their thoughtful and constructive feedback, which has been invaluable in improving our manuscript. Below, we address each of the reviewer’s concerns and outline the corresponding revisions.
>
> **Q1: Modeling Continuous Data Distribution is Not Sufficiently Challenging or Novel**
>
> We appreciate the reviewer’s comment on the perceived lack of novelty in modeling continuous data distributions and would like to further clarify our motivation and contributions:
> - **Continuous Data Modeling as a Challenging Problem**:
> The modeling of continuous data remains an active area of research, as evidenced by the significant advances in diffusion and flow-matching models. These models were originally designed for continuous data distributions and have achieved remarkable success before being adapted to discrete domains.
> - **Novelty of Our Approach**:
> CaLMFlow builds on the foundation of the flow matching framework, which is designed for modeling continuous data distributions, and extends it by uniquely adapting large language models (LLMs)—traditionally designed for discrete data such as text—to continuous data distributions. This adaptation is both non-trivial and meaningful, expanding the applicability of LLMs and advancing the field. By doing so, we contribute to a theoretical understanding of flow matching with Volterra integral equations, offering more robust modeling of stiff systems and enabling native support for natural language as conditions for generation.
>
> We have revised the manuscript to emphasize the novelty and significance of applying LLMs to continuous data modeling more explicitly.
>
> Please refer to Section 1 and 2 for the changes.
>
> **Q2: Gaussian and MNIST Experiments are Too Trivial**
>
> We thank the reviewer for this comment and would like to provide additional context for these experiments:
> 1. **Gaussian Experiments**:
> Gaussian distributions have often been used as synthetic examples in flow matching literature to provide intuitive demonstrations of model capabilities [1, 2, 3]. In our experiments, we intentionally design Gaussian scenarios that are simple yet challenging for ODE-based models due to their high dimensionality, which naturally introduces stiffness. These results underscore the limitations of ODE-based flow-matching methods in addressing stiff systems, even when the underlying dynamics seem straightforward, such as evolving Gaussians. This limitation motivates the development of our framework, which extends ODE-based flow matching to Volterra integral equations (VIEs), offering improved robustness and adaptability for high-dimensional, stiff dynamical systems.
> 2. **MNIST Experiments**:
> We acknowledge that MNIST is not a particularly challenging dataset; however, its inclusion provides a straightforward example to showcase the potential application of CaLMFlow in the vision domain. The primary focus of our real-world validation lies in the single-cell experiments. These experiments align with the common practices in recent flow matching [1, 2, 4, 5] and continuous normalizing flow [3] literature. Moreover, our single-cell experiments go beyond the scope of prior work by being more advanced and rigorous, making them well-suited for demonstrating the capabilities of our model. The objective of these experiments is to highlight CaLMFlow’s ability to:
>    - Model complex, high-dimensional data distributions.
>    - Generalize to out-of-distribution (OOD) data samples. (In our conditional generation experiment, CaLMFlow successfully generates data with combinatorial labels completely unseen during training.)
>    - Leverage natural language conditions intuitively and natively for generation.
>
> We are also preparing experiments with additional data modalities to demonstrate the broader applicability of our framework.

---

> ### Author Response · Authors · 2024-11-19
> **Official Response by the Authors [2/2]**
>
> **Q3: Overclaim of Novelty for the Tokenization Strategy**
>
> We appreciate the reviewer’s feedback. While we acknowledge that similar tokenization strategies may have been employed in other contexts, the novelty of our work lies in leveraging these strategies to enable the modeling of spatiotemporal dynamics via LLMs, rather than in the tokenization strategies themselves. Specifically, our tokenization approach facilitates the integration over spatiotemporal domains, allowing LLMs to effectively model complex dynamical systems using autoregressive language models.
>
> Furthermore, we extend these strategies to support the tokenization of multiple trajectories simultaneously, enabling the LLM to incorporate a larger context of trajectories. To the best of our knowledge, this application has not been explored in the context of flow matching models. Importantly, we view our tokenization strategy as a means to an end—our primary contribution lies in advancing flow matching using LLMs, which is the core focus of this work.
>
> We have revised the manuscript to clarify that tokenization is a supporting technique and not the primary novelty.
>
> Please refer to Section 4 for more details.
>
> **Q4: Problem Definition and Parameter Comparison**
>
> We thank the reviewer for pointing out the need for detailed experiment descriptions. We have expanded the relevant sections in the Appendix to provide clearer explanations. Here, we summarize:
> - **Synthetic Experiments**:
> The source and target distributions are generated dynamically during training until the model successfully captures the underlying dynamics. Training samples consist of sequences of trajectories, starting with a sample from the source distribution and ending with a sample in the target distribution. Intermediate points are obtained via linear interpolation.
> - **Fair Parameter Comparison**:
> For all experiments, we ensured that the number of parameters across models was kept roughly equal to maintain a fair comparison.  For the single-cell experiments, we have included tables listing the number of parameters of all models compared.
> Please see Appendix A for details.
>
> We hope these clarifications and revisions address the reviewer’s concerns. We thank the reviewer once again for their valuable comments, which have significantly strengthened our manuscript.
>
> **References**
>
> [1] A. Tong, K. Fatras, N. Malkin, G. Huguet, Y. Zhang, J. Rector-Brooks, G. Wolf, and Y. Bengio. Improving and generalizing flow-based generative models with minibatch optimal transport, 2024.
>
> [2] K. Kapu´sniak, P. Potaptchik, T. Reu, L. Zhang, A. Tong, M. Bronstein, A. J. Bose, and F. D. Giovanni. Metric flow matching for smooth interpolations on the data manifold, 2024.
>
> [3] A. Tong, J. Huang, G. Wolf, D. van Dijk, and S. Krishnaswamy. Trajectorynet: A dynamic optimal transport network for modeling cellular dynamics, 2020.
>
> [4] D. Haviv, A.-A. Pooladian, D. Pe’er, and B. Amos. Wasserstein flow matching: Generative modeling over families of distributions, 2024.
>
> [5] L. Atanackovic, X. Zhang, B. Amos, M. Blanchette, L. J. Lee, Y. Bengio, A. Tong, and K. Neklyudov. Meta flow matching: Integrating vector fields on the wasserstein manifold, 2024.

---

> > ### Comment · Reviewer_LGVE · 2024-11-26
> >
> > Thank you for the comprehensive feedback. While some of my questions have been addressed, I still have concerns about the triviality of the task, the novelty of the method (particularly the tokenizer), the misalignment between the motivation and the experiment (ODE vs continuous data modeling). Therefore, I will maintain my score.

---

> ### Author Response · Authors · 2024-11-26
>
> We thank the reviewer for the response.
>
> We would like to request some clarification regarding the reviewer’s comment here and in your review.
>
> >I still have concerns about the triviality of the task
>
> Could the reviewer please clarify what they find trivial about each of these tasks?
> 1. Our experiments show that traditional flow matching methods struggle on the Gaussian toy tasks despite being a simple data distribution.
> 2. Strong performance on generalization in perturbation response prediction remains a difficult task in single-cell transcriptomics [1]. While generative single-cell models exist, they are not effective on generalization tasks, as evidenced by our experiment. It would be beneficial if the reviewer can point to literature showing generative models with strong generalization performance on perturbation response prediction tasks in single-cell RNAseq data.
>
> >the novelty of the method (particularly the tokenizer)
>
> We kindly ask the reviewer to clarify and point to literature regarding the tokenizer and “standard window-based segmentation techniques”. We are not sure if the reviewer is referring to sliding window techniques, long-context methods that aggregate local information into single tokens to reduce context length, or something else entirely. Perhaps if the reviewer can specify similar work in the literature, we would be better able to compare our method.
>
> >the misalignment between the motivation and the experiment (ODE vs continuous data modeling).
> Could the reviewer clarify what is meant by “continuous data modeling”, which is referenced both here and in the original review?
>
> Video, image, dynamical systems, protein folding, single-cell, audio, and geophysical data are all highly non-trivial continuous data modalities. Furthermore, flow matching is a generative technique designed for continuous data, similar to diffusion. It is not entirely clear what the reviewer’s concern is about “continuous data modeling”, which is a rich and highly non-trivial subject in machine learning research.
>
> Re ODE methods:
> 1. Neural ODE methods, which include ODE-based flow matching, are also designed for continuous data [2]. Thus, it is not entirely clear what “ODE vs continuous data modeling” means.
> 2. We outlined in our rebuttal in the section on Gaussian experiments why we conducted those experiments, which were meant to highlight stiffness and high-dimensional issues with ODE solvers. Could the reviewer please clarify why there is a mismatch? Which experiments were conducted that did not correspond to our claims?
>
> **References**
>
> [1] Deep learning-based predictions of gene perturbation effects do not yet outperform simple linear methods, Constantin Ahlmann-Eltze, Wolfgang Huber, Simon Anders
> bioRxiv 2024.09.16.613342; doi: https://doi.org/10.1101/2024.09.16.613342
>
> [2] Chen, R. T. Q., Rubanova, Y., Bettencourt, J., & Duvenaud, D. K. (2018). Neural Ordinary Differential Equations. Advances in Neural Information Processing Systems, 31.

---

### Official Review · Reviewer_KKhb · 2024-11-04

**Soundness:** 3
**Presentation:** 2
**Contribution:** 3
**Rating:** 8
**Confidence:** 3

**Summary:**

The paper introduces CaLMFlow, a framework that leverages LLM for continuous data generation. The paper formulates flow matching as a sequence modeling task, and applies LLMs to learn the complex flows. By tokenizing the space and time, the approach enables efficient handling of high-dim data. On a range of tasks (e.g., single-cell perturbation response prediction), the method shows strong performance.

**Strengths:**

The technical idea is original and natural, blending LLMs into the framework of VIE flow matching.

Extensive experiments were carried out on a range of tasks, covering synthetic and real-world data.
The analysis and ablation studies also show the importance of each component/technique in the method, providing insights for future improvements.

The paper is generally clear.

**Weaknesses:**

The method seems general but the only kind of real data in the experiments are single-cell data. Without further evidence, it is hard to judge whether the significance will be high or not, broad or not.

The writing has some typesetting issues. E.g.,
- line-213, $T$ tokens
- line-219, $D_{text{in}}$

**Questions:**

What are the author's thoughts on generalizing the method to other kind of spatiotemporal data, e.g., high-dim time series, video data, etc?

---

> ### Author Response · Authors · 2024-11-19
> **Official Response by the Authors**
>
> We sincerely thank the reviewer for their insightful and constructive feedback on our manuscript. Below, we address the points raised regarding additional real-world experiments and the generalization of our approach to spatiotemporal data.
>
> **Additional Real-World Experiment**
>
> We are grateful to the reviewer for highlighting the importance of demonstrating the generalizability of our framework. We are actively working to prepare experiments with additional data modalities beyond single-cell data to showcase the broader applicability of our approach.
>
> Instead, we would like to emphasize the significance and relevance of our single-cell experiments:
> 1. Relevance of Single-Cell Data in Recent Research:
>    Single-cell data has been widely adopted as a real-world application in recent studies on flow matching [1, 2, 4, 5] and continuous normalizing flows [3]. This is because single-cell data inherently reflects complex biological dynamical processes, making it a natural choice for evaluating generative models.
> 2. Advancements in Our Experiments:
>    Compared to prior work, our single-cell experiments are both more advanced and extensive:
>    - Unconditional Generation: Demonstrates CaLMFlow’s ability to model high-dimensional, real-world, and complex distributions effectively.
>    - Conditional Generation: Highlights CaLMFlow’s capacity to generalize to out-of-distribution (OOD) data by generating samples with combinatorial labels that are entirely unseen during training.
>    - Integration with Natural Language Conditions: Unlike many models that struggle with encoding conditions, CaLMFlow leverages the inherent strengths of LLMs to support natural language conditions intuitively and natively.
>
> We believe these contributions demonstrate the robustness and versatility of our approach. We will revise the manuscript to emphasize these points more clearly and thoroughly.
>
> **Generalization to Spatiotemporal Data**
>
> We thank the reviewer for raising this intriguing point. Our framework is indeed naturally extensible to modeling spatiotemporal data.
>
> In this paper, we have focused on generating artificial spatiotemporal dynamics, where noise samples evolve into actual data points, demonstrating our framework’s effectiveness as a generative model. With suitable adjustments, we are confident that the framework can be extended to real-world spatiotemporal dynamical systems. Video data, for instance, represents an excellent example of such applications.
>
> We agree that exploring these applications is an exciting and promising future direction. However, given the scope of the current paper, we have limited our focus to foundational experiments and applications. We will add this discussion to the manuscript to outline potential avenues for extending our work.
>
> We thank the reviewer again for their valuable suggestions, which have helped us improve both the focus and clarity of our manuscript.
>
> **References**
>
> [1] A. Tong, K. Fatras, N. Malkin, G. Huguet, Y. Zhang, J. Rector-Brooks, G. Wolf, and Y. Bengio. Improving and generalizing flow-based generative models with minibatch optimal transport, 2024.
>
> [2] K. Kapu´sniak, P. Potaptchik, T. Reu, L. Zhang, A. Tong, M. Bronstein, A. J. Bose, and F. D. Giovanni. Metric flow matching for smooth interpolations on the data manifold, 2024.
>
> [3] A. Tong, J. Huang, G. Wolf, D. van Dijk, and S. Krishnaswamy. Trajectorynet: A dynamic optimal transport network for modeling cellular dynamics, 2020.
>
> [4] D. Haviv, A.-A. Pooladian, D. Pe’er, and B. Amos. Wasserstein flow matching: Generative modeling over families of distributions, 2024.
>
> [5] L. Atanackovic, X. Zhang, B. Amos, M. Blanchette, L. J. Lee, Y. Bengio, A. Tong, and K. Neklyudov. Meta flow matching: Integrating vector fields on the wasserstein manifold, 2024.

---

> > ### Comment · Reviewer_KKhb · 2024-11-26
> >
> > Thank you for your extensive responses.
> > I would like to maintain my current positive rating.

---

> > > ### Author Response · Authors · 2024-11-26
> > >
> > > Thank you for your insightful reviews and active engagement in the rebuttal process.

---

### Official Review · Reviewer_htMr · 2024-11-05

**Soundness:** 3
**Presentation:** 1
**Contribution:** 3
**Rating:** 3
**Confidence:** 3

**Summary:**

In this paper, the authors propose to use an autoregressive (language) model to imitate a (discretized) flow from the prior to the target distribution. This has a clear advantage, as such an autoregressive model with an unbounded context can take into account non-local structures across multiple timesteps, unlike previous approaches that model a purely local transition (a la “difference”). Once training is over (in the next-token prediction way), one can simply run the autoregressive model forward starting from a sample drawn from the prior distribution to draw a sample from the target distribution.

**Strengths:**

It is only natural to extend any of these iterative refinement based approaches to learning to sample from a complex distribution to make each refinement step less local and more global. Although it is natural, it has been challenging to do so until recently, as it was not clear whether we can build a powerful neural net that can take as input a long sequence of a trajectory and use it properly. With the recent advances in language models, this doubt is no more, and the authors in this paper demonstrate that indeed we can use such an autoregressive model to build a better sampler based on iterative refinement. Despite the unnecessarily convoluted way of presenting it via so-called Volterra flow, the idea is extremely straightforward, and the authors’ limited experiments do support that there is a benefit to be had from such non-local iterative refinement. It is furthermore interesting to see that they could easily extend it by prefixing their iterative refinement sampler with a natural language description to make it language conditional.

**Weaknesses:**

Unfortunately the current manuscript is extremely difficult to read. One reason i can point out is due to the lack of clear exposition on how this underlying autoregressive model looks like; what does it take as input, and how the prefix is processed to result in the prediction of the next step’s refined observation? Furthermore, the authors’ use of the term “tokenization” confused me quite a lot, as “tokenization” is often used to refer to as the process by which we quantize a continuous observation into a finite set of discrete tokens, while the authors here are referring to discretizing continuous time and space. Such issues with presentation make it pretty much impossible for me to recommend the manuscript in the current form to be accepted at the venue. I would suggest the authors make a major revision to present the proposed approach (which I think may have an interesting impact on the community) more clearly and thoroughly.

**Questions:**

This is more of a question to the other reviewers as much as it is to the authors. I am not too familiar with usual practices when these flow-based approaches are introduced and compared. Are these experiments standard ones that people have used to compare different iterative refinement based samplers? The first set of experiments on mixtures of Gaussians look extremely low-dimensional, and the second set of experiments on single cell sequencing data look relatively unconventional (I like this problem but it is difficult for me to understand that this is one of the standard problems people use to compare these samplers.) I may be completely wrong here (and if so, please do point out,) but I would like the authors to clarify what are more or less standard benchmarks in this area and how the proposed approach compares to the existing ones on those.

---

> ### Author Response · Authors · 2024-11-19
> **Official Response by the Authors [1/2]**
>
> We sincerely thank the reviewer for their thoughtful and constructive feedback on our manuscript. We appreciate the reviewer’s valuable insights into improving the readability and clarity of our manuscript. Upon reviewing the paper, we realized that there are some points not stated clearly enough. Accordingly, we have thoroughly revised the manuscript to address the issues raised. The updated version includes detailed explanations, improved organization, and clarifications. **Please refer to the updated manuscript (revised texts highlighted in blue)  for more details.** Below, we highlight the key changes made in response to the reviewer’s comments:
>
> **Q1: “What does this underlying autoregressive model look like, and what are its inputs and outputs?”**
>
> We utilized small (<10m parameters) custom GPT-2 architectures  [6] from OpenAI for the synthetic dataset experiments and  Pythia architectures  [7] from EleutherAI for the single-cell experiments. These models were chosen because they are lightweight and simple, fitting within the constraints of our limited compute resources, while still allowing us to focus on the contributions of our framework rather than the inherent capabilities of a large model, with Pythia  providing slightly more capacity for a real-world experiment. By avoiding the use of massive models like LLaMA-3, we aim to highlight that the success of our approach does not depend on the size of the underlying language model but rather on the innovations in our methodology. Both GPT-2 and Pythia are causal language models that process input sequences of tokens and output sequences of tokens using an attention mechanism, making them well-suited for our task.
>
> In our approach, we construct input data as follows:
> - A batch of data samples (e.g., Gaussian or single-cell data) is sampled.
> - We sample a corresponding batch of noise samples and interpolate between the two to generate a trajectory from noise to data. This results in a tensor of shape [batch_size, num_timesteps, data_dimension].
>
> Depending on the tokenization strategy used, the tensor may be transformed further:
> 1. Spatiotemporal Embedding: Data samples are split and projected into multiple tokens, producing a tensor of shape [batch_size, num_timesteps * num_space_tokens, projected_data_dimension].
> 2. Multi-Trajectory Embedding: Multiple trajectories are sampled and concatenated along the sequence dimension.
>
> The LLM performs causal attention over the sequence dimension, effectively modeling an integral equation. Further details can be found in the manuscript and references therein.
>
> Please see Section 4.1 for an improved explanation. See Appendix B for an algorithmic description.
>
> **Q2: “The use of the word tokenization is confusing.”**
>
> Thank you for pointing out the ambiguity in our use of the term “tokenization.” We acknowledge that this differs from its conventional use in natural language processing. In our context, tokenization refers to representing spatiotemporal dynamics as input sequences of embeddings for LLMs, enabling them to perform integration over the corresponding domains.
>
> We have revised the manuscript to clarify this distinction and have updated the terminology for greater precision and alignment with the intended meaning.
>
> Please refer to Section 4.1 for the revised version.
>
> **Q3: “How is the input prefix processed?”**
>
> Conditional generation from text prompts is a key innovation of our method. We appreciate the reviewer highlighting the need for clearer exposition of this process.
>
> To generate conditional inputs:
> 1. Conditions are expressed as text prompts.
> 2. These prompts are tokenized using the LLM’s tokenizer and embedded into vectors via the LLM’s embedding layer.
> 3. The resulting text embeddings are concatenated with the flow matching trajectories to form the input sequence.
>
> This process leverages LLMs' native capability for text understanding, making it more straightforward where for other models such processes are non-trivial.
>
> Please see Section A.1.3 of the Appendix for a concrete example.

---

> ### Author Response · Authors · 2024-11-19
> **Official Response by the Authors [2/2]**
>
> **Q4: Standardized Benchmark for Flow Matching Models**
>
> We appreciate the reviewer’s question regarding benchmarks. Flow matching is an emerging area of research that has been explored and developed across various domains, including single-cell analysis [1,2,3,4,5], protein design [8,9], and image processing [1]. While it's hard to find standardized benchmarks widely accepted by the whole community, single-cell experiments serve as a crucial benchmark and have been utilized in prior studies [1,2,3,4,5] for its high dimensional property and practical value. We mainly build on this single cell benchmark, and include some image benchmarks as auxiliary (as shown in Table 6 and Figure 9). But thank you again for pointing it out! We also plan to extend our approach to additional data modalities in future work.
>
> We want to also take the opportunity to emphasize the significance of our single-cell experiments here. Our single-cell experiments rigorously test flow matching models in a challenging, high-dimensional setting. Specifically, CaLMFlow demonstrates robust modeling of complex biological processes in single-cell data and excels at out-of-distribution (OOD) generalization. For example, in our conditional generation tasks, CaLMFlow successfully generates combinatorial labels that are completely unseen during training, outperforming methods like scVI.
>
> We have revised the manuscript to highlight the following:
> - Our alignment with existing benchmarks in recent literature.
> - The significant contribution of CaLMFlow in advancing flow matching for practical applications.
> - The practical value of our approach, particularly in handling challenging real-world datasets like single-cell data.
>
> We hope these revisions address the reviewer’s concerns and enhance the manuscript’s clarity and impact.
>
> We are deeply grateful for the reviewer’s thoughtful feedback, which has significantly improved the quality and clarity of our work. Thank you again for your time and effort in reviewing our manuscript.
>
> **References**
>
> [1] A. Tong, K. Fatras, N. Malkin, G. Huguet, Y. Zhang, J. Rector-Brooks, G. Wolf, and Y. Bengio. Improving and generalizing flow-based generative models with minibatch optimal transport, 2024.
>
> [2] K. Kapu´sniak, P. Potaptchik, T. Reu, L. Zhang, A. Tong, M. Bronstein, A. J. Bose, and F. D. Giovanni. Metric flow matching for smooth interpolations on the data manifold, 2024.
>
> [3] A. Tong, J. Huang, G. Wolf, D. van Dijk, and S. Krishnaswamy. Trajectorynet: A dynamic optimal transport network for modeling cellular dynamics, 2020.
>
> [4] D. Haviv, A.-A. Pooladian, D. Pe’er, and B. Amos. Wasserstein flow matching: Generative modeling over families of distributions, 2024.
>
> [5] L. Atanackovic, X. Zhang, B. Amos, M. Blanchette, L. J. Lee, Y. Bengio, A. Tong, and K. Neklyudov. Meta flow matching: Integrating vector fields on the wasserstein manifold, 2024.
>
> [6] A. Radford, J. Wu, R. Child, D. Luan, D. Amodei, and I. Sutskever. Language models are unsupervised multitask learners. 2019.
>
> [7] S. Biderman, H. Schoelkopf, Q. Anthony, H. Bradley, K. O’Brien, E. Hallahan, M. A. Khan, S. Purohit, U. S. Prashanth, E. Raff, A. Skowron, L. Sutawika, and O. van der Wal. Pythia: A suite for analyzing large language models across training and scaling, 2023.
>
> [8] J. Yim, A. Campbell, A. Y. K. Foong, M. Gastegger, J. Jim´enez-Luna, S. Lewis, V. G. Satorras, B. S. Veeling, R. Barzilay, T. Jaakkola, and F. No´e. Fast protein backbone generation with se(3) flow matching, 2023
>
> [9] A. J. Bose, T. Akhound-Sadegh, G. Huguet, K. Fatras, J. Rector-Brooks, C.-H. Liu, A. C. Nica, M. Korablyov, M. Bronstein, and A. Tong. Se(3)-stochastic flow matching for protein backbone generation, 2024.

---

> ### Author Response · Authors · 2024-12-02
>
> We would like to thank the reviewer for their valuable feedback. We have revised the manuscript to make the language and our approach clearer and provided context for our choice of benchmarks. Since this is the final day for reviewer comments, we would greatly appreciate the reviewer's additional thoughts on our revisions.

---

### Author Response · Authors · 2024-11-22

We would like to thank the reviewers for their comments and suggestions for the manuscript. With only a few days left in the discussion period, we would be grateful to hear your thoughts and responses to our comments, answers, and revised manuscript.

---

### Meta-Review · Area_Chair_CR7P · 2024-12-19

**Metareview:**

The paper introduces CaLMFlow (Causal Language Models for Flow Matching), a novel framework that reformulates flow matching as a Volterra integral equation (VIE) (instead of a differential equation), leveraging large language models (LLMs) for continuous data generation. The authors claim that CaLMFlow bridges discrete language modeling and continuous generative modeling by implementing tokenization across space and time, enabling efficient handling of high-dimensional data. The method is demonstrated on synthetic and a real-world data application (single cell sequencing dynamic), showcasing its ability to incorporate textual context and generalize to unseen conditions.

**Strength: Timely, Innovative Connection of two Active Areas of Research**. There is a long line of work aiming to repurpose neural network architectures typically used in language modeling (RNNs, LSTMs, Transformers) for continuous data domains. This paper fits withing that broader literatures, approaching this problem by marrying LLMs and flow matching, two very active areas of research, but so far mostly independent. Therein lies, in my view, the main strength of the paper: an innovative approach that presents a unique application of LLMs to flow matching, bringing the promise of flow matching to other domains. The use of integral (instead of differential) equations to model flow matching is interesting, but not an original contribution of this work (see Zappala et al.). Furthermore, I think its use requires more motivation. The paper makes a broad argument that IEs should be preferred over ODEs, but there's inherent computational-robustness tradeoffs that are not discussed here in depth. Additionally, the paper implicitly assumes that the conditions under which IEs might be preferable (singularities, non-locality, weak solutions) are ipso facto true in the applications of interest, but this does not seem to be case in the synthetic Gaussian experiments (despite their high-dimensionality), and unclear in the single cell perturbation ones. A stronger case for VIEs would have been made with an application where ODEs provably suffer from such issues, or to demonstrate that the decreased performance of ODE-based CFM in table 1 is due to stiffness or instability and not implementation issues.

**Weakness: Methodological choices lack motivation (see above)**.

**Weakness: Clarity and Presentation**. The reviewers noted that the manuscript is difficult to read due to unclear exposition and confusing terminology, particularly around the use of “tokenization.”. The revised version made important improvements in clarity, but introduced substantial changes. While such improvements are welcome, the fact that such an extensive rewriting was necessary suggests that the paper was submitted in a form not ready for prime time. Furthermore, while the revised version is definitely easier to read, many of the points raised by reviewer ktMR about missing details still stand.

**Weakness: experimental evaluation**. As observed by all three reviewers, the experimental framework consists of very simplistic synthetic datasets and non-standard (for flow matching) single-cell generation task. This makes it hard to compare the method to alternatives. Additionally, there does not seem to be experimental results comparing the computational cost of this method against the baselines. This is particularly problematic given the known robustness-computation tradeoff of solving integral vs differential equations.

Overall, I think this is a borderline paper with high unrealized potential that would benefit from a broader evaluation, rewriting, and an additional round of peer review.

**Additional Comments On Reviewer Discussion:**

* Reviewer htMr's main concerns focused on writing and presentation issues, in addition to the simplicity of the experimental framework. I agree on both fronts. They did not respond to the author's response.
* Reviewer KKhb was appreciative of the novelty of the method and certain aspects of the evaluation. However, they too noted the simplicity of the experiments and the need for broader validation in other types of spatio-temporal data.
* Reviewer LGVe raised concerns about the 'triviality' of the tasks, the novelty of the tokenization method, and the alignment between the motivation and experiments. Although some of these concerns could have been better expressed, I tend to agree with the substance of some of them. For example, I do know that spatiotemporal tokenization has already been introduced (e.g., in the CV literature [1]), which makes the claim of contribution on that aspect feel exaggerated. This reviewer stated that many of their concerns had not been addressed by the rebuttal.




[1] Wen et al., "Interactive Spatiotemporal Token Attention Network for Skeleton-based General Interactive Action Recognition".

---

### Decision · Program_Chairs · 2025-01-22

Reject